# Reprogramming of regulatory network using expression uncovers sex-specific gene regulation in *Drosophila*

Yijie Wang[1], Dong-Yeon Cho[1], Hangnoh Lee[2], Justin Fear[2], Brian Oliver[2] & Teresa M. Przytycka [1]

Gene regulatory networks (GRNs) describe regulatory relationships between transcription factors (TFs) and their target genes. Computational methods to infer GRNs typically combine evidence across different conditions to infer context-agnostic networks. We develop a method, Network Reprogramming using EXpression (NetREX), that constructs a context-specific GRN given context-specific expression data and a context-agnostic prior network. NetREX remodels the prior network to obtain the topology that provides the best explanation for expression data. Because NetREX utilizes prior network topology, we also develop PriorBoost, a method that evaluates a prior network in terms of its consistency with the expression data. We validate NetREX and PriorBoost using the "gold standard" *E. coli* GRN from the DREAM5 network inference challenge and apply them to construct sex-specific *Drosophila* GRNs. NetREX constructed sex-specific *Drosophila* GRNs that, on all applied measures, outperform networks obtained from other methods indicating that NetREX is an important milestone toward building more accurate GRNs.

[1] National Center of Biotechnology Information, National Library of Medicine, NIH, Bethesda, MD 20894, USA. [2] Laboratory of Cellular and Developmental Biology, National Institute of Diabetes and Digestive and Kidney Diseases, 50 South Drive, Bethesda, MD 20892, USA. These authors contributed equally: Yijie Wang, Dong-Yeon Cho.  Correspondence and requests for materials should be addressed to B.O. (email: briano@niddk.nih.gov) or to T.M.P. (email: przytyck@ncbi.nlm.nih.gov)

Maintenance of cell type-specific states, response to stress, sexual dimorphism, and other cell functions are controlled by gene regulatory programs. In particular, gene regulatory networks (GRNs) capture the regulatory relationships between transcription factors (TFs) and their target genes. Since GRNs provide information that is essential for a global understanding of the logic of gene–gene interactions, inference of these networks is one of the key challenges in system biology. Methods to infer GRNs typically combine computational approaches and experimental data collected from different sample types, different conditions, different techniques, and different labs. Such data integration leverages dependencies that can be confidently uncovered thanks to the multitude of surveyed conditions, but leads to context-agnostic wiring diagrams[1–3]. These context-agnostic networks do not accommodate regulatory program reality, which is specific to tissue types, developmental stages, sex, and other factors.

To study tissue, developmental stage, or sex-specific gene regulation, context-specific regulatory networks are needed. *Drosophila* sex differentiation is an ideal test for such context-dependent models, as sexual dimorphism results in subtle differences in every germ layer and tissue[4]. Thus, models of sex-biased expression will show many differences between the sexes, but also a core of gene regulatory relationships that should be similar between the sexes. The most readily accessible context-specific data type is context-specific gene expression. Therefore a spectrum of methods to construct GRNs from only gene expression data have been developed, counting on the relation between expression of TFs and expression of their target genes. In recent years, many methods that infer GRNs based on gene expression alone have been proposed. Early methods inferred regulatory relationships using mutual information between the expression levels of gene pairs[5,6]. These approaches have been followed by more sophisticated ones that account for more complex regulatory scenarios[7–12]. The recent DREAM5 network inference challenge[13] evaluated over 30 expression-based network inference methods and identified a random forest-based method, GENIE3, as the best performer. However the results of this challenge demonstrated that expression only methods are far from solving the GRN network inference problem suggesting that relying on expression only is not enough. One of the factors that led to the limited success of these methods is the complicated relationship between expression of TFs and their regulatory activity[14], indicating that it might be beneficial to rely on the TF regulatory activities inferred from the data rather than TF expression per se. For example, network component analysis (NCA) has been shown to be a successful approach to infer such regulatory activities[15]. Unfortunately, NCA requires prior knowledge of the GRN in order to infer TF activities but, in our setting, the GRN is largely unknown. As a result of this difficulty, effort has been extended to integrate prior knowledge from different types of experiments, or even from different conditions, to provide additional ways to boost inference of such networks[16–21]. For example, the Inferelator[21] method uses a prior network in place of a true network as the input to the NCA procedure to infer TF activities, and then predicts a GRN based on relationships between the inferred TF activities and gene expression[21].

Here we introduce, NetREX, a method to construct GRNs by iterative reprogramming of a prior network, given a prior network and expression data. In applications to predict context-specific GRNs, the prior network is assumed to reflect a prior information that might not be context specific, while the expression data provide the context. NetREX can be applied to any situation where a prior network is to be improved by expression data. The main idea of NetREX is to reprogram the prior network by adding and removing edges to obtain a network that provides the best explanation of the observed gene expression. Simultaneously, NetREX optimizes several other objectives to ensure that the resulting network is biologically relevant. NetREX is an approach that systematically explores the landscape of possible GRN topologies to generate context-specific GRNs.

NetREX, and all other models that use a prior, assume that there is some similarity/overlap between the prior network and the target GRN, and thus these tools bias the optimization procedure toward networks that overlap with the prior. Therefore, in the case of significant discrepancies between the prior and the target network, the prior might be misleading rather than helpful. To address this challenge we developed PriorBoost—a computational approach to gauge the usefulness of the prior network for obtaining a good estimation of the target GRN.

We validated NetREX and PriorBoost—first on simulated data and then on the "gold standard" *E. coli* GRN used in the DREAM5[13] challenge. As an additional evaluation, we compare how well the methods predicted novel regulatory edges that have been added to the *E. coli* RegulonDB[22] after the DREAM5 challenge. NetREX outperforms other methods on different metrics. Additionally, PriorBoost successfully identifies priors that are likely to lead to misleading results.

We then apply NetREX and PriorBoost to construct sex-specific GRNs for adult *Drosophila melanogaster* using a previously constructed context-agnostic network as the prior[2]. We supply a large expression dataset for adult female and male flies where perturbations in expression were achieved by heterozygosity for multi-locus deletions[23,24] to NetREX to generate the sex-specific GRNs. We evaluate the performance by evaluating the subnetwork centered on the sex-specific transcription factor Doublesex (*DSX*), which is the key gene controlling, directly or indirectly, the majority of sex differentiation in *Drosophila*[25]. *DSX* occupancy in *D. melanogaster*, and the comparative genomics of *DSX* binding motifs in the *Drosophila* genus have been extensively mapped to provide a good test of connectivity predicted by NetREX. Furthermore, we illustrate that, among all competing methods, only *DSX* targets predicted by NetREX are enriched in genes with sex-biased expression. Finally, we demonstrate that while GRNs inferred by NetREX show differences between the sexes, their regulatory programs overlapped, consistent with the similarities between the sexes.

## Results

**NetREX and PriorBoost overview.** The main idea of NetREX is to construct a context-specific GRN by leveraging an existing GRN—for example a GRN constructed in a related tissue or organism, or a noisy/incomplete network for the same context. The context of interest is provided by a set of expression data. NetREX edits the prior network by removing and adding edges to obtain a network topology that provides the best explanation for the entirety of the expression data. To accomplish this, NetREX requires four components: (i) a measure of how well a network topology explains the expression data, (ii) a strategy for exploring biologically relevant network topologies, (iii) an algorithmic technique guaranteeing convergence of the network search procedure, and (iv) a method to test whether the given prior is consistent with the data and likely to provide an advance over prior-free methods. Below we provide basic intuition underlying these four components. The details of the method and its mathematical underpinning are described in Methods section.

To measure how well a given network's topology explains the expression data, we needed to have a mathematical model linking network topology to gene expression. NetREX uses the network component analysis (NCA) model[26] (Supplementary Figure 1), which assumes that each TF is characterized by its activity (TF

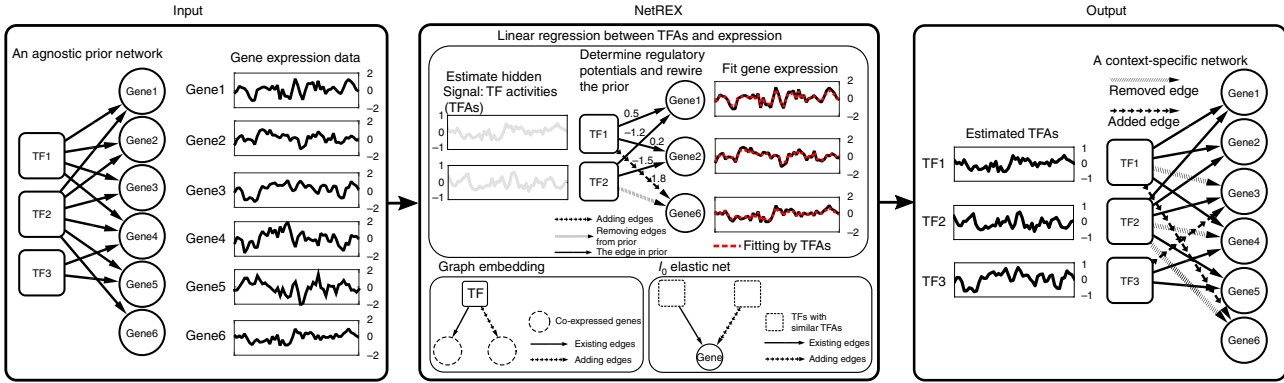

**Fig. 1** Schematic outline of NetREX using a simulated example with 3 TFs, 6 target genes and their expression measurements

activity), a variable that is not directly measured but introduced to account for unknown factors, such as protein levels, nuclear localization, and phosphorylation status. In addition, in the NCA model, each edge of the GRN has a weight representing regulatory potential (or strength) with which the TF regulates the gene. Finally, the expression of a gene is assumed to be a linear combination of the activities of TFs that regulate the gene, weighted by regulatory potentials of the regulatory edges (Supplementary Figure 1). To measure how well network's topology explains the expression data, NetREX measures the fitness of the consistency of the given topology with the expression data using the optimal NCA model. Despite the fact that this model is relatively simple (Discussion), we verified the efficacy showing that computed explanatory power correlates with the number of "gold standard" edges in the E. coli GRN (Supplementary Figure 2), motivating our use of this metrics as a measure of the relationship of network topology to expression data.

Starting from the prior NetREX iteratively reprograms it by adding and removing edges giving preferences to topologies where co-expressed genes are coregulated and TFs with correlated activities coregulate the same genes (Fig. 1 and Methods) and penalizing the number of changes (see Methods and Supplementary Methods: The Formulation of NetREX)

Computationally, NetREX is formulated as an optimization problem with $l_0$ norm involved, making the problem non-convex and NP hard. We addressed this challenge by using a new cutting edge technique known as proximal alternative linearized maximization (PALM)[27] as described in Supplementary Methods: Optimization Behind the NetREX Algorithm.

NetREX is a prior-based method, and therefore performance critically depends on the prior. To avoid erroneous solutions due to a poor prior, we developed PriorBoost, to evaluate the usefulness of a prior network for the task of reconstructing a GRN consistent with a given expression dataset (Methods).

**Benchmarking NetREX**. While benchmarking against a true network is ideal, no current GRNs are perfect. Therefore, we first tested the performance of NetREX on simulated data. Overall NetREX solution provided a consistent improvement over the initial prior and the improvement increased with less noise in the expression and/or a higher fraction of true positive edges in the prior (Supplementary Figure 3).

Next, to see how the method can handle a situation of non-random error in the prior network we simulated the scenario where the prior is consistent with the true network in most cases except one truly differential module of genes. NetREX performed very well even in the case when all true edges leading to the

module have been removed from the prior (Supplementary Figure 4).

Complementing benchmarking the method on simulated data, we evaluated NetREX on currently the most complete GRN[22], the E. coli network. Following the strategy used in the DREAM5 challenge[4], we used the same experimentally validated high-confidence interactions from the curated dataset RegulonDB[22] as a reasonable "gold standard" set and the same expression data that was provided to the DREAM5 competitors. We evaluated the ability of NetREX to recover this "gold standard" network as a function of the quality of the prior. As in the case of simulated data, we constructed prior networks of various quality by randomly selecting a subset of edges from the "gold standard" network as true positives and randomly adding false positive edges. We compared NetREX with Inferelator[21], MERLIN+P[20], and CoRegNet[7], all of which use a prior network (see parameters selection in Supplementary Note 6). In addition, we included Genie3[11]—the best performer in the DREAM5 challenge that uses expression data only (no prior). We varied the difficulty of the network inference problem by using prior networks generated in two ways. The first set of noisy prior networks had the same number of total edges, but different percentages of true edges. The second set of noisy priors had the same number of true edges, but different numbers of total edges which are controlled by the ratio of true to false edges. We assessed the quality of the predicted networks by AUPR (the Area Under the Precision vs. Recall curve) scores. The results using AUROC (Area Under the Receiver Operator Characteristics curve) are similar and provided in (Supplementary Tables 1 and 3). Except for the case when the prior network contained only 10% of true edges (Fig. 2a–c) and no true edges (ratio of true to false edges is 0:1 in Fig. 2d–f), NetREX outperformed all other methods under most test conditions. Genie3 outperformed all other methods when the prior network contained very low percentage of true edges (Fig. 2a, b, d, e), which is consistent with the expectation that if the prior is a poor match, the algorithms not using that prior gain an advantage. Performance of MERLIN+P was overall not significantly influenced by the quality of a prior and close to the performance of GENIE3. Interestingly NetREX was the only method that provided a consistent improvement over the provided prior (curves of NetREX in Fig. 2a, d are always above the curves of the prior). When the prior contained >60% correct edges, the network constructed by Inferelator's was actually worse than the prior network provided as the input. In this aspect, we attribute the superior performance of NetREX in part to the fact that it gives preference to the solutions that are close to the prior. We also tested the impact of sample size on method's performance. NetREX provides improvement over the prior with as little as 10 samples and the performance continues to steeply

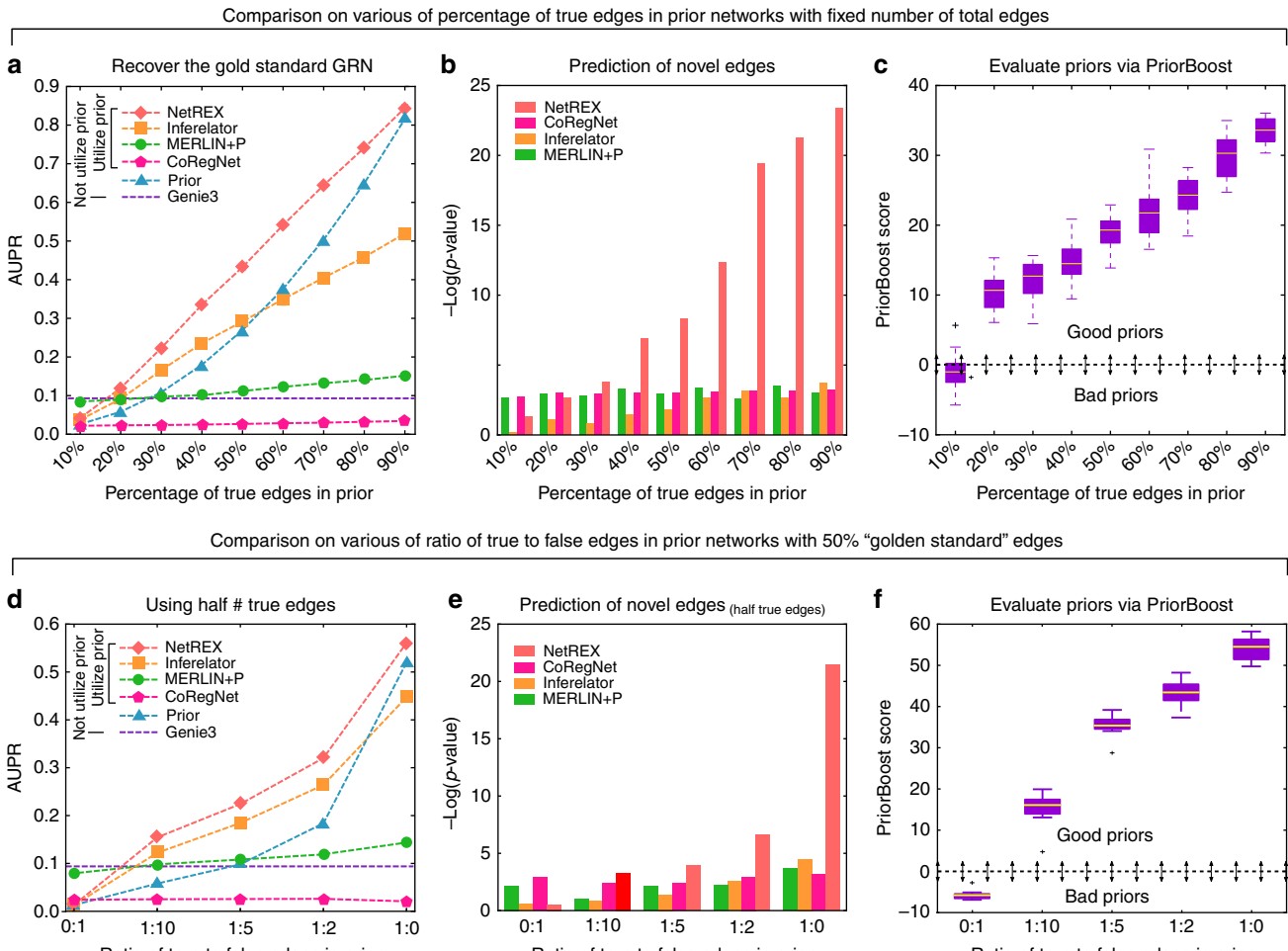

**Fig. 2** Comparison of network inference methods based on *E. coli* data. **a** The performance measured in terms of AUPR as a function of the percentage of true edges in the prior. The total number of edges in the prior networks is fixed and equal to the number of edges in the "gold standard" set. **b** Recovery of the novel TF-gene interactions (*x*-axis same as **a**). **c** PriorBoost scores (*x*-axis same as **a**). For each box in the boxplot, the central mark (white line) indicates the median, and the bottom and top edges of the box indicate the 25th and 75th percentiles, respectively. **d**–**f** Same as **a**–**c** but when the total number of true edges is fixed and equal to half of the number of "gold standard" edges

increase with sample size and plateaus around 100 samples (Supplementary Figure 7).

Even if the *E. coli* GRN is currently the most complete network, it is not perfect. Therefore we performed additional validations, using new data. Specifically, we acquired 230 novel high-confidence interactions from RegulonDB[22] (Methods) that we added to the dataset RegulonDB[22] after the DREAM5 challenge was completed, and thus not included in the "gold standard" set. We then tested whether those novel edges were uncovered by competing algorithms (using the -log(p-value) from hypergeo-metric test that is used to compute the enrichment of novel edges in the set of total novel edges found by the algorithms). Again, except for the case of the lowest quality prior (CoRegNet has the best performance in predicting novel edges for the lowest quality prior), NetREX outperformed other methods (see Supplementary Tables 5 and 6).

Finally, we used *E. coli* network to validate our PriorBoost scoring system. Due to the dependence on the prior, NetREX, or any other model that uses a prior, could be mislead by a prior that is mostly wrong. This is observed in Fig. 2, where when the prior network had 90% false negatives (the very left points in Fig. 2a), both NetREX and Inferelator perform badly. To evaluate the prior network in the absence of "gold standard" truth, PriorBoost applies the above described theoretical model on *E. coli* data

(given expression and priors). Figure 2c and f shows the robustness of PriorBoost scores for the perturbed prior networks (used by NetREX, Inferelator, MERLIN+P, and CoRegNet for Fig. 2a, b, d, e) with different noise levels. As demonstrated in Fig. 2c, f, PriorBoost scores correlate with the quality of the prior. In addition, a negative PriorBoost score correctly identified a situation when NetREX cannot improve over Genie3.

**Reconstruction of *Drosophila* sex-specific GRNs.** We applied NetREX and PriorBoost to construct sex-specific female and male GRNs for *Drosophila*. The adult female and male gene expression data were obtained from a large collection of expression profiles (99 lines of flies, with females and males profiled separately in replicates) that were perturbed by altering gene dose[23,24]. This dataset provides a relatively large number of related samples that also have broad variability in gene expression patterns due to gene dosage alteration. Specifically, the dataset is derived from engineered chromosomal deletions each of which leads to dele-tion of one of the two copies of a block of genes from different regions. Because all these deletions are heterozygous (viable and fertile in this state), there are not secondary (and worse) effects due to defects in development. All the flies are morphologically wild type. As demonstrated in refs. [28,29] the expression changes

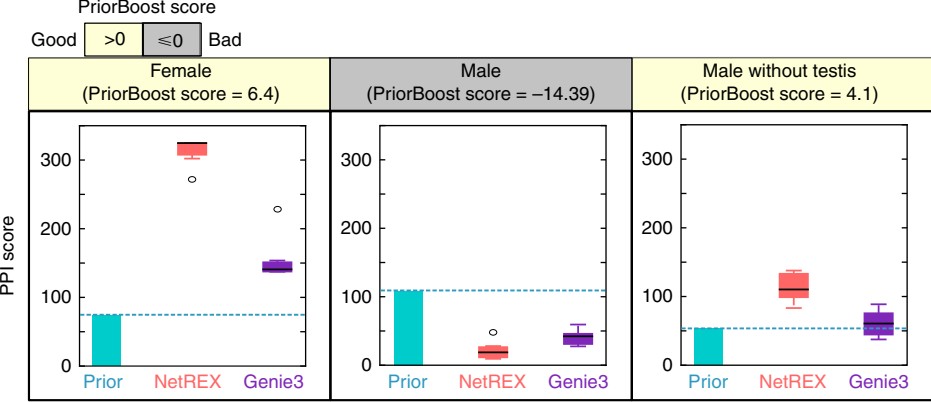

**Fig. 3** Estimating the usefulness of the prior network for the task of inferring sex-specific GRNs for *Drosophila*. The relevance of a prior network is estimated by PriorBoost scores where negative score indicate that the prior might be misleading. As an additional test we used PPI score which evaluates the topology of a network independently of its relation to expression data. When the PriorBoost scores are positive, PPI scores of NetREX are higher than GENIE3. For each box in the boxplot, the central mark (black line) indicates the median, and the bottom and top edges of the box indicate the 25th and 75th percentiles, respectively

caused by these genetic perturbations propagate and dissipate in gene network space, making this an ideal set for expression-based network reconstruction. Specifically, while transcriptional effects are perturbed, the underlying GRN is unbroken. These significantly perturbed expression profiles explore the expression space for the whole genome, as collectively essentially all genes show differential expression in at least one deletion. In addition, our estimates suggests that this set of ~100 experiments per sex (each in two biological replicates) is a sufficiently large dataset for NetREX to perform exceptionally well (Supplementary Note 4). For the prior network, we used a previously constructed conext-agnostic network[2]. This network was constructed through integrating diverse functional genomics datasets in a supervised learning framework. Since much of the evidence used for the construction of this network was based on experiments performed on tissue culture cells, which shows significantly different expression patterns relative to sexed adult flies, it was clear that extensive rewiring would be required to constructing adult sex-specific networks. The prior networks for female and male are basically the same and correspond to the network predicted in refs. [30,31]. However, genes that were not expressed were removed from the prior. Since the set of non-expressed genes in females and males is not exactly the same, this introduces a subtle difference between the two priors (Supplementary Table 7). To test the validity of using this prior for adult sex-specific networks, we first used PriorBoost to test the consistency of the prior GRN with female and male gene expression data. PriorBoost score was positive for female expression data indicating an informative prior, but was low for the male data (Fig. 3).

As an indirect way to evaluate the topology of a network, we used protein–protein interaction (PPI) scores and gene ontology (GO) scores (Methods)[2]. Starting with the assumption that coregulated genes are more likely to belong to the same pathway, these scores measure enrichment in PPIs and consistency of GO annotations of coregulated genes. While these scores do not measure correctness of the network, they provide a coherency estimate to determine whether the network topology has expected network properties. We revised these scoring functions relative to their original definition (Methods) and show, using the *E. coli* network, which revised scores have improved correlation with network quality (Supplementary Figure 5). We used these scores to gauge the quality of NetREX and Genie3 networks under the same cutoffs (e.g., top 50,000, 100,000, 150,000… weighted edges). Consistent with PriorBoost scores, the networks produced by NetREX had very high scores for the female networks but relatively low scores for male networks (Fig. 3 for PPI scores and Supplementary Tables 8–10 for GO scores).

The good performance for females was gratifying, but the poor performance of the prior for males was unsurprising, as several lines of evidence indicate that the organizational principles of the regulatory program of the testis is unique[32,33–43]. The *Drosophila* testis has a radically different gene expression machinery compared to any other tissue[32,33,35]. There are probably several causes of this special gene expression profile. Given that little of this unique "TF free" expression program (see Supplementary Note 3 for further discussion of this issue) was represented in the prior, this was a reassuring test for PriorBoost. If the poor performance of the prior for the male-specific GRN was indeed due to the peculiar nature of testis gene expression, then removing testis-biased expression should improve the prior performance. Indeed, the PriorBoost score for the prior network of the remaining genes was positive, and thus we used this network as a prior for reconstructing a male-specific GRN without genes highly expressed in testis. The resulting network showed also a good performance, as measured by PPI scores (Fig. 3). In the remaining analysis, to avoid any bias, we did not include genes highly expressed in testis (for males) or ovary (for female). The female-specific and male-specific GRNs constructed by NetREX are provided in Supplementary Data 1 and Supplementary Data 2.

To validate the resulting GRNs, we measured the overlap of the predicted targets of the key transcription factor for controlling the majority of sex-biased expression in flies, doublesex (*DSX*), with the identified targets from a combination of occupancy, binding motif, and comparative genomics[25]. Neither *DSX* occupancy, nor *DSX* binding sites were included in the prior. The expression data resulting from direct perturbation of *DSX* activity was not used either. Since the prior network is based largely on embryos and tissue culture cells, not surprisingly, it contained only three of the thousands of predicted *DSX* targets. Therefore the performance of the method on predicting the *DSX* targets is particularly informative. NetREX was able to identify, with high precision, hundreds of these independently verified target genes (Fig. 4a, b). In particular, the top 100 NetREX predictions had 72 verified targets (the highest of all sets listed in Fig. 4c) as compared to MERLIN+P and Genie3 that predicted 52 and 66 verified targets

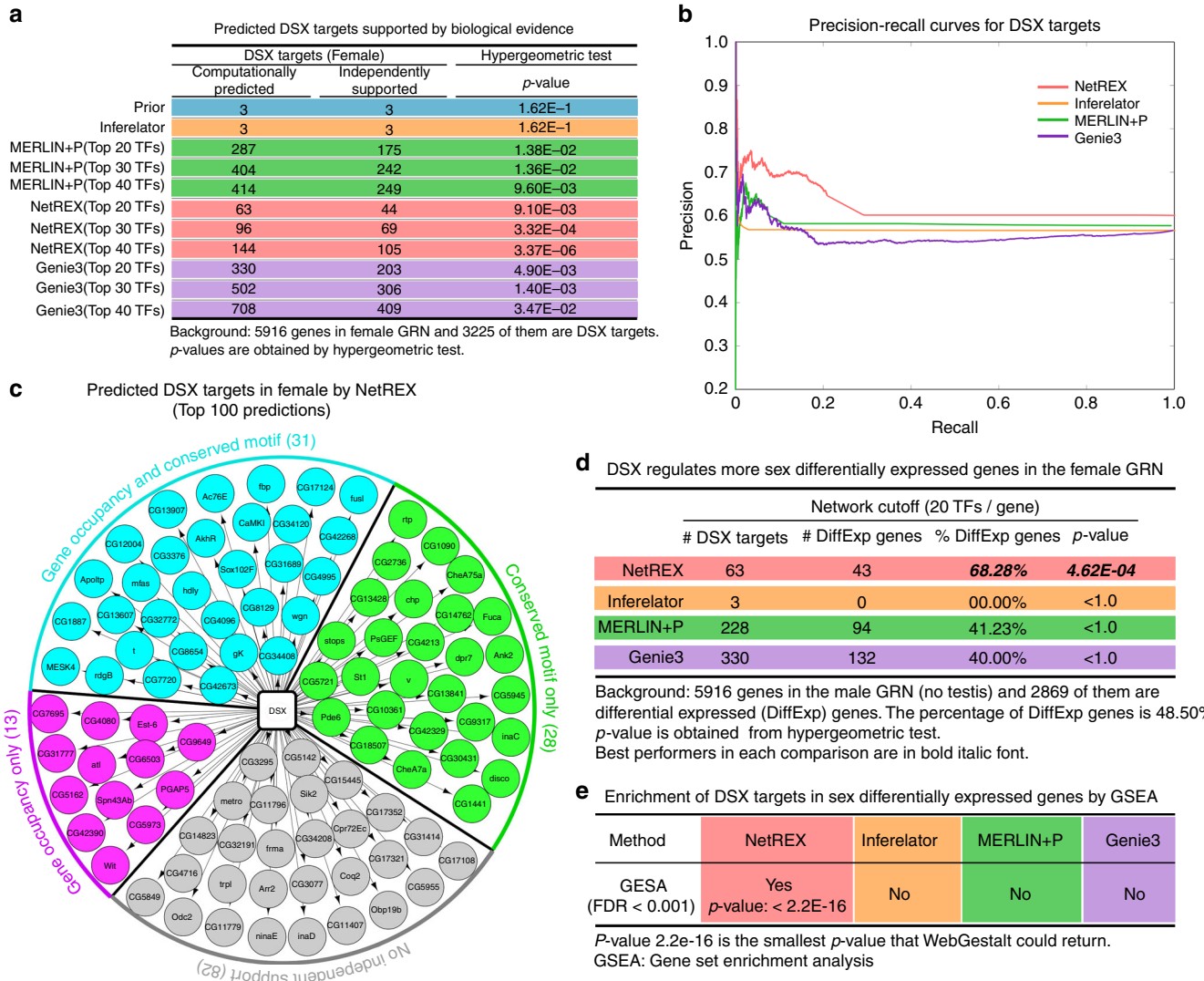

**Fig. 4** Validation of predicted DXS targets (predicted without genes highly expressed in ovary). **a** Enrichment of experimentally supported *DSX* targets recovered by different methods for female GRN. Enrichment for male GRN without testis is shown in Supplementary Table 11. **b** Precision-recall curves for predicting *DSX* targets for compared methods. The *DSX* targets predicted by each method are ranked by assigned weights. A high area under the curve corresponds to high precision (low false positive rate) and high recall (low false negative rate). As the ground truth we use *DSX* targets reported in ref. [30] based on ChiP-Seq occupancy and conserved motif scores. **c** Top 100 targets predicted by NetREX in the female GRN. **d** Enrichment of predicted *DSX* targets in genes with sex-biased expression for the female GRN. Different methods might predict different number of regulators for each gene. To fairly compare those GRNs we take for each method the *k*-best (*k* = 20) predictions for each gene. Comparison of other *k*s is shown in Supplementary Figure 8b. **e** Enrichment of *DSX* targets in sex differentially expressed genes by GSEA (Gene Set Enrichment Analysis)

in their top 100 predictions, respectively. Inferotaltor inferred only three interactions. Overall, NetREX clearly outperformed other approaches on this test.

For an additional validation, we utilized the fact that, since *DSX* controls sexual development, the targets of *DSX* are expected to be enriched in genes that are differentially expressed between females and males, even though not all *DSX* targets are sex-specifically expressed at any given time in development[25]. To test for the enrichment, and to avoid any confounding due to using the same expression dataset used to generate the network models for the validation, we obtained a second dataset of sex-biased expression from GEO Series accession number GSE99574 (96 samples from GSM2647254 to GSM2647349) and used it to identify genes with sex-biased expression (details in

Supplementary Note 4). When we asked what genes were predicted to be *DSX* targets in the predicted GRNs, we found that there were significantly more genes with sex-biased expression[44] among those predictions in the NetREX models (hypergeometric test in Fig. 4d; gene set enrichment analysis in Fig. 4e). The other tested GRNs failed to show a significant enrichment for sex-biased gene expression among the predicted *DSX* targets. These data indicated that NetREX can successfully predict gene expression patterns in a novel experimental dataset.

As yet another test, we evaluated similarities between the female and male GRNs. There are 505,548 and 293,458 edges predicted by NetREX for female and male GRNs. We found that 149,462 edges are common to the female and male GRNs. Of these, 136,404 are included in the prior and 13,058 edges were

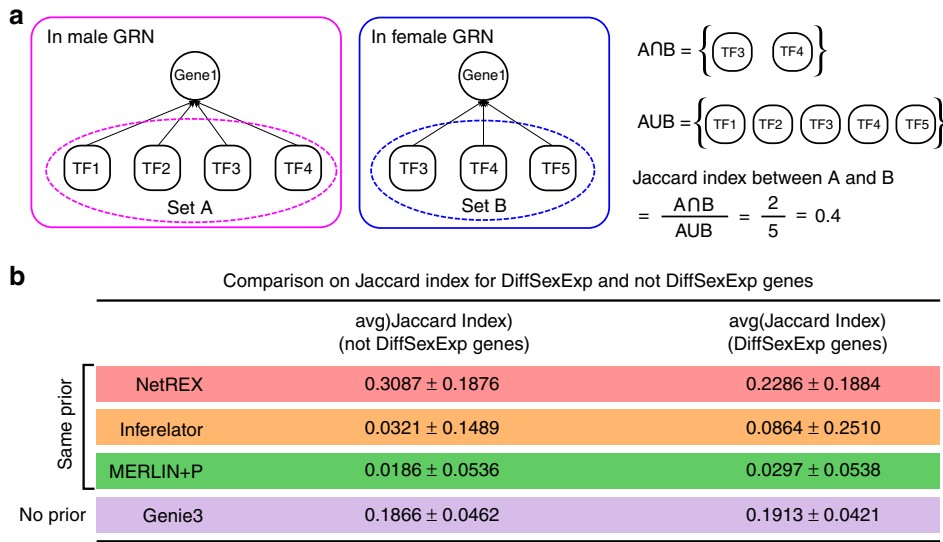

**Fig. 5** Evaluation of similarities of female and male GRNs. **a** Illustration of the method to compute the similarity of regulatory program in female and male GRNs for a fixed gene (Jaccard index). **b** Testing the similarity of the inferred regulatory programs in female and male GRN using all predicted edges. DiffSexExp genes are defined using independent data (details in Supplementary Note 4). The spread of the Jaccard indexes is their standard deviation

newly predicted. While many differences between the GRNs exist, these networks are expected to be related, as there is also much in common between female and male adult *Drosophila*, and there are many genes that do not show, or show only modest, sex-biased expression. We measured the similarity of regulatory programs by comparing for each gene the agreement between TFs predicted to regulate that gene in the female and male GRNs (Fig. 5a). Thus, we separately evaluated consistency of regulatory programs on sex-biased gene expression and on non-sex-biased expression (Fig. 5b). Female and male GRNs inferred by NetREX show overall good consistency between their regulatory programs (Jaccard index above 0.2 in all sets) (Fig. 5b). This is in contrast to the other methods where the average intersection/union (Jaccard indexes) in all the tests are much smaller. Thus, NetREX shows an outstanding improvement in identifying common aspects of gene expression among the sexes. Furthermore, accounting for imperfections in the GRN network prediction, we still expect that genes that are not sex differentially expressed between male and female have higher similarity of regulatory interactions than genes that show sex-specific expression. This is indeed what we found in Fig. 5b. The average Jaccard index for non-sex differentially expressed genes are much larger than the average Jaccard index for sex differentially expressed genes.

## Discussion

Gene regulation is context dependent. Gene regulatory networks depend on tissue, sex, developmental stages, and disease status among many other conditions. Ultimately, every cell type at any given time has a slightly different network than spatially or temporally neighboring cells. Clearly, universal network models will fail to capture this complexity. But, capturing this regulatory complexity is essential for elucidating the differences between regulatory networks in healthy and disease states, during development, and essentially any other biological condition. Thus, context-specific models are fundamental for understanding global regulatory mechanisms. However, direct measurement and modeling of context-dependent GRNs is a tremendous challenge, as a human, for example, is composed of roughly 37 trillion cells[45]. Despite advances in single cell genomics, inferring GRNs for each organism/tissue/cell/condition separately through

accumulation of huge numbers of condition-specific measurements is, and is likely to remain, impractical. We need methods that can leverage a smaller number of prior networks. For example a reference GRN for *Drosophila melanogaster* might provide information about GRNs for related species, and wild-type models of specific *Drosophila* tissues and stages might inform the changes that occur when those networks are perturbed by mutations and/or environmental conditions. In this way, a context-agnostic network provides a good first approximation prior for the wiring diagram of a context-specific network that explains developmental progression or disease. Gene expression information is currently one of the most easily accessible context-specific data types. Therefore, it is important to be able to utilize this data, along with the prior knowledge, for construction of context-specific GRNs. To address this need, we introduced here a GRN inference method—NetREX. The unique property of NetREX is that starting from a prior network, it utilizes expression data to interactively remodel a new network that converges on the observed expression patterns by adding and removing edges. The fact that NetREX explores the network space around a prior network gives us a unique advantage when the target network is at least marginally similar to the prior network. The evaluation of the method on *E.coli* network suggested that NetREX outperforms other methods when the overlap between prior and target network is ~20%. In addition it is the only method that continues to improve over the prior network even when this network is already quite good. In addition, NetREX performs very well on novel experimental datasets both in terms of predicting independently validated interactions and in terms of network consistency. For all these reasons, NetREX is a significant milestone in development of context-dependent network models from a limited set of adaptable reference networks.

While poor network models might someday be rare, currently many prior networks will have such poor quality that rewriting is futile, or can even degrade the performance of the model. In those cases, one would want to start from a model that does not use a prior. In the last decade, a significant effort has been devoted to prior-less construction of GRNs from gene expression alone. Thus, it is important to have a method to evaluate the trade-off between prior-based and prior-free approaches. To address this challenge, we introduced PriorBoost; a method allowing a

researcher to quantitatively gauge whether a given prior is helpful in the context of constructing a context-specific network model from a given expression dataset. We demonstrated that Prior-Boost was valuable for evaluating the context-agnostic networks as a possible prior for constructing both *E. coli*, and sex-specific *Drosophila* GRNs, and used this method to show that a prior for males was inappropriate when unusual testis-specific expression was used. By using PriorBoost we could eliminate *ad hoc* decision-making on the utility of prior models.

Several new methodological advancements introduced in this work contributed to the success of NetREX. These contributions include the design of the objective function which, in addition to evaluating the fit of the network, favors a network search toward biologically relevant topologies. However, since adding and removing of edges proceeds in discrete steps, the function optimized by NetREX is not continuous. Typical way of dealing with this issue is to convert the function to a continuous one (in this case by replacing $l_0$ norm which is discrete by $l_1$ norm which is continuous) and use standard optimization techniques on so modified problem even if it is not equivalent to the original one. An additional contribution of this work is the development of mathematical underpinnings allowing us to guarantee convergence of NetREX search by utilizing the cutting edge PALM optimization framework[27]. The applicability of these algorithmic advances, especially convergence of calculations that include $l_0$ norm, has broader applications to diverse feature selection approaches[46,47].

A key feature of NetREX is the ability to score the quality of network topology given expression data, in absence of the ground truth. For this purpose, NetREX utilizes the NCA model. This model is based on the assumption that gene expression can be modeled as a linear combination of activities of regulating TF, which is an oversimplification, but might approach the truth. Engineered gene expression modules in *Drosophila* show that TFs and enhancers act in a largely additive fashion as simple input/output devices[48].

While it was remarkable that, in the case of *E. coli*, NetREX was able to improve over a network that was that already 80% or more correct, the ultimate test for a GRN is to use it to make biological predictions. We not only used NetREX to construct the first sex-specific regulatory networks for *Drosophila*, but we demonstrated that NetREX outperformed networks obtained with alternative methods. For example, NetREX identified *Darkener of abricot* (*Doa*) locus as female target of *DSX*. The *Doa* locus encodes a kinase that is a positive feedback regulator of the *DSX* pre-mRNA splicing event that generates female-specific *DSX* TF[49,50]. We also provide methods to avoid inappropriate application of NetREX. PriorBost allowed us to directly determine whether a prior was suitable for rewriting a context-agnostic network, as demonstrated for accommodating unusual testis gene expression regulation due to specialized basal transcriptional machinery.

Overall our results show that NetREX is a very powerful method for integrating prior knowledge and expression data for reconstructing context-specific GRNs. While NetREX strongly relies on the initial prior, however by utilizing introduced here PriorBoost technique, it avoids using an inappropriate prior and being mislead by it.

## Methods

**NetREX.** In contrast to most of the previous methods that rely on the predictive power of the mRNA level of the TF (which might not reflect the cellular activity of the TF[51]), NetREX reconstructs a GRN based on unknown TF activities $A$. NetREX simultaneously estimates unknown TF activities $A$ and rewires the prior network $G_0$ until the structure of the rewired network $S$ and the predicted TF activities $A$ optimally explain the context-specific expression data $E$ based on the linear relationship described as $E(i,:) = \sum_j S(i,j) \times A(j,:) + \Gamma(i,:)$, where $E(i,:)$ represents expression of gene $i$, $S(i,j)$ represents the interaction between TF $j$ and gene $i$ and

its regulatory potential, $A(j,:)$ is the TF activity of TF $j$, and $\Gamma(i,:)$ represents the noise. Therefore, NetREX is formulated as an optimization problem (1) that aims to find the optimal linear model with several additional terms controlled by $\lambda, \kappa, \eta, \xi, \mu$ designed to enforce important properties of the target regulatory network as described below.

$$\min_{S,A} \tfrac{1}{2}\|E - SA\|_F^2 + \lambda\left(\|S_0\|_0 - \|S_0 \odot S\|_0 + \|\bar{S}_0 \odot S\|_0\right) + \kappa\mathrm{tr}(S^T L S)$$
$$+ \eta\|S_0\|_0 + \xi\|S\|_F^2 + \mu\|A\|_F^2 \tag{1}$$
$$s.t. \|S\|_\infty \le a, \|A\|_\infty \le b.$$

where $S$ is the adjacency matrix of network $G$ that is the output of NetREX. $\|\cdot\|_0$, $\|\cdot\|_F$, and $\|\cdot\|_\infty$ are $l_0$, Frobenius, and infinity norms, respectively. The $\|\cdot\|_0$ norm computes the number of non-zero elements in the matrix of interest. More mathematical details about the formulation can be found in Supplementary Methods.

The term controlled by $\lambda$ restricts the number of edge changes from the prior network (Supplementary Methods: The Formulation of NetREX). Larger $\lambda$ indicates that only small number of edges can be added and removed controlling how far our predicted network $G$ is from the prior network $G_0$. The term controlled by $\kappa$ (the graph embedding term[52]) encourages related genes encoded in gene–gene network $G^E$ to be coregulated by the same TFs (Supplementary Methods: The Formulation of NetREX). Here $G^E$ is the gene correlation network constructed based on gene expression data $E$ and $L$ is the corresponding *Laplacian* matrix. The terms controlled by parameters $\eta$ and $\xi$, which we call the $l_0$ elastic net, encourage sparsity of the final network with group effect (Supplementary Methods: The Formulation of NetREX). For the reader familiar with the elastic net model, we point out that the $l_0$ elastic net is analogous to $l_1$ elastic net[53]. Notably, the graph embedding and $l_0$ elastic net only encourages edges with certain property but does not remove edges. NetREX only removes edges if it finds TFs whose activities can better explain the expression of gene(s) other than the TFs in the prior network. Finally, the last term controlled by the variable $\mu$ enforces smoothness of activities in $A$ by limiting the number of elements in $A$ reaching the limit $\{-b,b\}$. The strategy of selecting parameters for NetREX is discussed in (Supplementary Note 6).

The optimization problem (1) with given parameters can be solved by using the Proximal Alternative Linearized Maximization (PALM) algorithm[27] which guarantees convergence (Supplementary Methods: Optimization Behind the NetREX Algorithm). The output of the PALM algorithm, $A$ and $S$, are the estimated TF activities and the predicted context-specific GRN, respectively. We can rank the edges in $S$ by their confidence score $B$ that measures their impacts on the overall performance of the linear model[18] (Supplementary Methods: Ranking Interactions and Bootstrapping).

$$B(i,j) = 1 - \frac{\left\|E(i,:) - \sum_{k \ne j} S(i,k)A(k,:)\right\|_F^2}{\|E(i,:) - S(i,:)A\|_F^2}. \tag{2}$$

To further improve the inference and make it more robust against overfitting and sampling errors, we use a bootstrapping strategy, where we resample the gene expression data with replacement and solve the problem (1) on the new dataset. This procedure is repeated several times, and the resulting lists of edges are combined to a final ranked list as in ref.[54]. For reconstruction of GRNs in a new context, where we do not have any ground truth information, different parameters are applied and the final ranking of the edges are obtained by consensus over the results under different parameters[54] (Supplementary Methods: Model Selection of NetREX). Parameter settings of NetREX for all experiments are elaborated in (Supplementary Note 6).

Efficiency and scalability are important for utility. NetREX needs to store the expression data and the prior network, therefore, the space complexity of NetREX is $O(NL+NM)$, where $N$ is the number of genes, $L$ is the number of samples, and $M$ is the number of TFs. Based on Algorithm 1 (Supplementary Methods: Optimization Behind the NetREX Algorithm), the heaviest computation in each iteration of NetREX is to compute the partial derivatives of the objective function, indicating that the time complexity of NetREX in each iteration is $O(NML)$. Therefore, the overall time complexity of NetREX is $O(CNML)$, where $C$ is the number of iterations that NetREX takes in a run. Both the space and time complexities scale linearly with respect to the number of samples $L$.

**PriorBoost.** The assessment of the prior network suitability is based on two ideas. First, the quality of any network $G$ can be estimated by the consistency between the structure of the network and the expression data. Such consistency is validated in *E. coli* data (Supplementary Methods: The PriorBoost Score and Supplementary Figure 2) and can be computed by the following equation.

$$q(G) := \min_{S \in G, A} \|E - SA\|_F^2. \tag{3}$$

$S \in G$ means that the non-zero pattern of $S$ is conserved to the structure induced by $G$. Actually, equation (3) is the original formulation of NCA[26] and $q(G)$ is the

optimal objective function value after solving NCA. Second, if a prior network is consistent with the given expression data, the network predicted by a prior-based method should be better than the network inferred by an expression-based method. The expression-based method we used here is Genie3, which was the winner of the DREAM4[54] and DREAM5[13] challenges.

Specifically, suppose we have a prior network $G_0$ and expression data $E$. $G^*$ is the network predicted using both expression $E$ and the prior $G_0$, and $\bar{G}$ is the network predicted by Genie3 using only expression $E$. Let $G_c^*$ and $\bar{G}_c$ be networks obtained by keeping the top $c$ edges in $G^*$ and $\bar{G}$ based on their edge weights, respectively. Then, the PriorBoost score for the prior network $G_0$ can be estimated by

$$Q(G_0) := \frac{1}{|C|} \sum_{c \in C} q(\bar{G}_c) - q(G_c^*), \tag{4}$$

where $C$ is a set of different cutoffs. Positive $Q(G_0)$ indicates that the network predicted using $E$ and $G_0$ is more consistent with the expression data $E$ than the network predicted by Genie3. A positive $Q(G_0)$ also implies that the prior network is informative, while a negative $Q(G_0)$ indicates the opposite.

**Novel TF-gene interactions for _E. coli_.** In addition to the 2066 TF-gene interactions used in DREAM5 challenge, we identified 230 additional interactions that were discovered after DREAM5 from RegulonDB 9.2 (version 09-08-2016)[22]. We utilized these 230 interactions to test the ability of each method to predict novel interactions.

**The PPI score.** One way to validate a GRN is to test whether physically interacting genes are preferentially coregulated. Here we introduce and validate a modification of the previously proposed score based on this idea[2]. We consider two genes are coregulated if the Jaccard similarity coefficient between the TF set regulating the first gene and the TF set regulating the second gene is >0.5. The Jaccard similarity coefficient between two sets is the ratio of the size of the intersection of the given two sets to the size of the union of these two sets. Our measure is based on the following hypergeometric test. Suppose that there are $N$ PPIs among $M$ gene pairs, and there are $m$ coregulated gene pairs in the predicted network with $n$ having PPIs. The p-value is the probability of selecting more than $n$ PPIs when we choose $m$ gene pairs at random. The PPI score is defined as $-\log_{10}(p\text{-value})$. We tested the PPI scores on simulated _E. coli_ GRN with different noise levels that are controlled by the percentage of true edges and the ratio of true to false edges. We found that the PPI score defined in this way are more consistent with the quality of the network compared to the previously proposed measure[2] (Supplementary Figure 5).

While PPI score can be very useful, it should be used with caution. In particular it should not be used to compare networks that are sparse (a network has to have a significant number of coregulated genes for the score to be meaningful) and, as any p-value-based score, it should not be used for comparing networks of very different sizes.

Finally, note that the PPI score is independent of expression data and thus it can be used to evaluate topology of the network but not its relation to the experimental data.

**The GO score.** The GO score of coregulated genes was computed analogously to the PPI score[2] with the following modification. For each coregulated gene pair, we again use the Jaccard similarity coefficient to measure the similarity between the GO annotation set corresponding to the first gene and the set corresponding to the second gene and consider the coregulated genes are functional similar if the similarity is >0.5. Instead of using all GO terms[55], we only considered high-level GO terms with information content (IC) larger than two so that we can better understand the functional specificity of the coregulated gene pairs[55,56]. The IC of a GO term $g$ is defined as $-\ln(|g|/|\text{root}|)$, where 'root' is the corresponding root GO term (either F, P, or C) of $g$[55,56]. We also used the hypergeometric test to get a p-value indicating the enrichment level of the functional similar gene pairs within the coregulated gene pairs inferred by the networks. The GO score is also defined as $-\log_{10}(p\text{-value})$. We illustrated the effectiveness of GO scores on simulated _E. coli_ GRNs (Supplementary Figure 5).

As in the case of PPI scores, computing GO scores might not be meaningful in some situations.

**The DSX targets.** The experimentally supported _DSX_ target genes are the union of two sets. The first set of genes were obtained based on ChIP-Seq gene level occupancy scores[25]. And the second set was collected based on conserved motif scores[25]. The experimentally supported _DSX_ target gene set was served as the ground truth for investigating the predictive power of different methods (details are in Supplementary Note 4).

**Highly expressed genes in ovary or testis.** We used the quantification of tissue-specific expression from modENCODE as summarized in FlyBase[57]. Flybase assigns genes to bins depending on their expression in a given tissue. "Bin_value" is an integer that ranges from 0 to 6, where 0 means that a gene has very low expression and 6 means it has extremely high expression. We identified all genes expressed in ovary or testis with "Bin_value" >5 and treat them as genes highly expressed in ovary or testis.

**Code availability**. The integrative networks, input and validation datasets, as well as the source code used for network inference and validation are provided in online supplementary information and on the companion website of the paper (https://www.ncbi.nlm.nih.gov/CBBresearch/Przytycka/index.cgi#netrex (Matlab) and https://github.com/ncbi/NetREX (Python)).

## Data availability

All the data used in this study (data for _E. coli_, female, and male flies) are included in https://www.ncbi.nlm.nih.gov/CBBresearch/Przytycka/index.cgi#netrex. And the female-specific and male-specific GRNs constructed by NetREX are provided in Supplementary Data 1 and Supplementary Data 2.

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

## Acknowledgements

We thank members of the Przytycka and Oliver labs for useful discussions. This research was supported by the Intramural Research Programs of the National Library of Medicine (NLM), National Institute of Diabetes and Digestive and Kidney Diseases (NIDDK), and the Korean Visiting Scientist Training Award (KVSTA, HI13C1282 to H.L.).

## Author contributions

Y.W., D.C., H.L. and J.F. conceived the research project. B.O. and T.P.M. supervised the research project. Y.W. and D.C. designed the computational formulation and algorithm. Y.W. implemented NetREX and tested it on the benchmark data. Y.W., D.C., H.L., J.F., B.O. and T.P.M. design evaluation methods to validate the predicted female and male fly GRNs. Y.W. applied the evaluation methods to analyse the results. Y.W., B.O. and T.P.M. wrote the manuscript with support from all the authors.

## Additional information

**Competing interests:** The authors declare no competing interests.

