## [Peer Review File · Nature Communications]

Reviewers' comments:

Reviewer #1 (Remarks to the Author):

The paper explains in a clear way an interesting approach, NetREX, to infer context-specific gene regulatory networks (GRNs) given context-specific gene expression data and an "incomplete" prior GRN. Additionally, a second method is presented, PriorBoost, to assess the informative power of a prior network. The use of statistical learning and graph theory techniques seems very promising. The theoretical foundation of the approach has been carefully designed and carried out, as further explained in the supplemental material.

They provided several benchmarks for their algorithm, and justified the choice of proposed metrics through informative tests. A real effort at interpretation of the results, and cross-checking with other data sources, has been realised. I have some suggestions to support the value of this method.

- It could be interesting to compare the results obtained with NetREX-GENIE3 with existing network tools that i) infer bipartite network (transcription factors regulate their targets) ii) can generate network-based latent representation of transcription factor activity. A few Bioconductor packages such as RTN (Fletcher Et al., 2013; Nature Communications) and CoRegNet (Nicolle et al. 2015; Bioinformatics) as well as others not named here should be included in the comparison and/or discussion. As the prior network is an important part of the NetRex workflow, it could be interesting to study the robustness of PriorBoost using for example a partially permuted version of the prior network with increasing levels of noise.

- The authors should think about availability of an open repository. Also, it would be better to provide a generic implementation of NetRex. The actual version is configured with the demo data and needs Matlab installed. A more detailed explanation should be given about the running time and whether the tool can handle large-scale datasets from RNA-seq and/or Human single cell data (20K cells vs 20K genes). Without an easy-to-use tool this application would be of low significance to the community.

Minor comments

- abstract : "We validated the predicted GRNs computationally and experimentally ... » This sentence may be confusing as there is no new experiments in the paper to support hypothesis generation. It looks more like cross-checking with other data sources

- Why choosing the Drosophila and E. Coli DREAM 5 data as they are now quite old datasets and better quality datasets can be obtained now with a better confidence on the differentially expressed genes.

- Perhaps consider changing the purple color in figures like Figure 4a by a lighter one (i.e. more readable)

- Some typos in the manuscript :

page 2: seuxal -> sexual

page 4: we demonstrated that the while (remove "the")

page 7: that infers (add "a") GRN based on

page 7: benefit to starting from -> benefit to start from

page 10: worse that -> worse than

page 11: to test (add "the") consistency

page 12: stores -> scores

page 12: NetREX (add "are") higher than
page 16: details (add "can") be found
page 16: a context-agnostic networks -> a context-agnostic network
page 17: NetREX (add "is") not continuous
page 19: for nonzero elements -> of nonzero elements

Supplementary - There are a lot of typos and some sentences that should be changed in order to be understood.

Reviewer #2 (Remarks to the Author):

Wang et al develop a novel computational approach to derive gene regulatory networks called NetREX. The approach uses prior networks and adds and removes edges to obtain a network that best explains the gene expression data. They also develop a computational approach called PriorBoost that determines if the prior network is useful in estimating gene regulatory networks. The authors validate their approach using simulated data and a "gold standard" gene regulatory network from *E. coli*. They next used their approaches to construct sex-specific gene regulatory networks from *Drosophila* data. The approach assumes that a transcription factors is characterized by its activities and gene regulation is due to a linear combination of the activities of transcription factors. The NetREX algorithm starts with a prior and then adds and removes edges, with preference given to network topologies that predict co-expressed genes are co-regulated and transcription factors with similar activities co-regulate the same genes.

To benchmark their new approaches, they first test using simulated data, where a "ground truth" is known. They show that their approach improved network prediction, by examining various levels of "noise" in their simulated expression data. Next they show using the "gold standard" *E. coli* data that NetREX outperforms other approaches that use priors, if the quality of the prior is good, with sufficient numbers of true edges. The authors next test their approach on *Drosophila* data, from adult animals harboring different deletion mutations. They generate a prior network based on several different experimental and computational data sets. The PriorBoost approach finds that the prior networks are not a good match for the male data that includes expression data from the germline, but that the prior network is useful on the female data. They proceed removing the testis gene expression and observe good performance. They also test the performance on a separate gene expression data and examine network predictions on non-sex-biased gene expression.

Overall, the benchmarking approaches show that NetREX outperforms other algorithms that predict gene regulatory networks, when there is sufficient prior information. The predicted DSX gene regulatory network from the female data set were significantly supported by the independent biological evidence, providing further proof-of-principle for this new network prediction approach. While the ability to improve prediction of gene networks is a significant step, some questions remain about the data sets used to validate NetREX that the authors need to address.

For the authors to address:

It is not clear why the authors used non-melanogaster data sets for the "second dataset of sex-biased expression". The GEO number given is GSE80124 and contains Hawaiian *Drosophila* species data. It is also not clear which data from this data set were analyzed. Given that DSX targets might be different in different species, more rationales as to why they chose a non-melanogaster data set are needed.

Given that DSX does not regulate gene expression in female germline tissues, does that impact the outcome of the gene regulatory networks constructed, especially when considering sex-biased genes? Including the female germline data and not the male germline data is likely to bias the genes that are called sex-biased in this study. How did the authors deal with this difference in

male and female data sets? Did the authors make gene regulatory networks from male and female data sets that are only from somatic tissues? If not, why were those not included?

The section regarding how non-sex-specific gene regulatory networks were predicted is confusing. What was the prior? Was this also based on the DSX priors? Can the authors provide evidence that there are large sets of DSX targets without sex differences in expression? What tissue/species were the gene expression data sets made from that were used here? There is too much ambiguity in this section to understand what is being presented.

Minor comments:

The use of the deletion strains is confusing. Can the authors provide more information about why those will show expression variation?

There are typos in the figures: Figure 1 should be signal and not "singal". Figure 2-I assume you mean gold standard, not "golden standard". There are also minor typos throughout the text.

The authors mention the top 100 NetREX predictions in 4A, but those are not presented in 4A.

Reviewer #3 (Remarks to the Author):

Summary comments:

This paper describes a method for building context-specific gene regulatory networks given a general specific prior network. It also describes a metric for determining the match between a prior network and a dataset of gene expression data. The authors further define a metric for the quality of a learned network based on PPI networks. Building context-specific networks is very important to the understanding of biological systems specifically in understanding differences between diseased and healthy tissues. Current methods either assume a prior network that was built independent of context or build a network from scratch using data from the system of interest. Using prior networks is an efficient way to both gain confidence in the inferred networks as well as decrease the amount of data required to infer new networks. The metrics provided by this paper for understanding the quality of a prior network and comparing a network to known biology are also important advancements in the field. The authors compare to other methods: two that use priors and another that learns the network directly from the expression data.

While the method seems conceptually interesting, the manuscript could benefit from increased clarity in description of the method in the main text as well as the statistical tests performed in the validation. The authors also should explicitly state the time or space complexity. They are using a method, PALM (proposed proximal alternative linearized maximization), to solve the optimization problem, whose complexity should be stated.

Major points:

- In figure 1, it seems as though edges between TF's and putative targets will be removed if the one TF is not co-expressed with the other TF's regulating that target. It is possible for a TF to be regulating a gene in an orthogonal way to the other TF's regulating it such that the TF would not be correlated with the other TF's regulating this gene. The authors should clarify if this is an assumption that the method makes.

- There is no discussion of the efficiency of the algorithm- is this feasible/fast to run? What is its space complexity?

- There is also limited discussion of how much expression data is required for the reported performance. Since expression data is expensive to acquire, it would be useful to have an understanding of the amount needed to obtain the quality reported for this method.

- It is unclear how NetREX quantifies statistical significance. The fact that it finds >100 predictions of DSX targets while these other networks get much fewer raises questions about the sensitivity of the method. How is it so much more powerful? (page 13 second to last paragraph).

- It would be useful if the authors specifically addressed how this method performs in the case that the prior is consistent with the true network in most cases except one truly differential module of genes. It seems like this method takes advantage of redundancy in biological networks; what happens if some subset of the network is incorrect in a concerted way? This could be very important in inferring networks for disease when using a healthy tissue as a prior and would be of great interest to know if the method handles this type of difference.

Specific points:

The authors allude to the fact that the genie3 network is sometimes closer to the truth than the prior. It would be interesting to see the performance when the genie3 network from the same expression data is used as a prior on NetREX.

1. Specific comments on the tests of the prediction of novel edges from RegulonDB:

- Were the new interactions they tested from regulonDB all in one module? Were they at all context specific? Some information about which edges these were and which ones were recovered could strengthen the argument that NetREX is specifically adept at identifying context-specific networks.

- In figure 2b, the authors should clarify how they calculated the p-value and what it is quantifying. Based on the supplementary tables (5 and 6) it seems that the hypergeometric test is testing for enrichment of novel edges in the total number of edges found by the algorithm, but these bar plots could also be interpreted as the enrichment of novel edges found by the algorithm in the set of total novel edges. A clarification of this difference would be useful to include.

- Tables 5/6 in the supplement indicate that there is a large difference in the total number of predicted edges between the three algorithms. While the AUPR metric addresses the question of false positive edges, including raw numbers on true vs. false positives in tables 5 and 6 as well as a comment on the differences between the numbers of edges found would add clarity.

- The column names in tables 5 and 6 ("# avg. novel" and "# avg. overall") should be explained more clearly (i.e. how many tests went into this average? What is the spread of the average? Are the same novel and overall edges found in each test of the same algorithm?)

- Referencing the section of the supplementary information (tables 5 and 6) containing the results of the test of recovering E.coli new interactions on page 10 in the second to last paragraph would increase readability.

2. Specific comments on recovering DSX targets from Drosophila data:

- While interesting, the description of the motivation to remove testes-specific genes might be off topic for this paper (bottom of page 12-page 13).

- The authors should clarify if the DSX targets are expected to be different for males and females and the degree to which they are different. Specifically, when reporting the results for enrichment of DSX targets identified for males and females, the authors should indicate how many of the identified targets were truly context-specific and how many were targets found in both male and female networks. The authors mention that the prior networks used for male and female were different (supplementary table 7), but in supplementary note section 2.3.2 the authors specify that the same prior network was used for both. The authors should clarify which network was used for each step. For the steps using different priors, the source of these networks should be clarified as well as the motivation for using different networks. The authors should also clarify which of these TF-target pairs are included in the prior. Is there a difference in the number of edges NetREX must add/remove from the prior for the male and female networks? Are the priors closer to the male or female context specific network? In supplementary section 2.3.5 the authors state that the NetREX recovers more overlaps than other methods, and knowing the number of these overlaps that were

included in the priors would help in understanding the advantages of this method.

- The methods' description for highly expressed genes in testis is vague. Where did the testis-specific gene list come from?
- The hypergeometric tests in figure 4a are unclear. The background should be specified more clearly as the interpretation of the p-value of 1 for the prior and the Inferralator method comparisons is confusing without specification of how the test was run. In supplementary table 11, the same values for predicted and verified genes as in figure 4a are listed for the prior network, but a different p-value is stated. A description of the background would help clarify this difference.
- When testing for the enrichment of DSX targets in differentially expressed genes, it is conventional to use Gene Set Enrichment Analysis as it takes into account the rankings of the differential expression of the genes. The authors should justify why the hypergeometric test is sufficient for this.
- The authors should clarify what the cutoff '30 TFs based on edge weights means' is referring to in supplement section 2.3.4 and in the sections beyond that.
- For the DESeq2 outputs in supplementary section 2.3.4, the authors state that the filtering for differentially expressed genes is done via a log2fold change cutoff. Usually, a cutoff for adjusted p-values is performed on the outputs of DESeq2 as well, does performing this cutoff change the outputs significantly?
- Some indication of the spread of the Jaccard index should be included for the averages in figure 5.

Typos:

- On page 2 in the top paragraph, the prose "accommodate regulatory program reality" is ambiguous
- on page 2 in the second paragraph there should be no comma after 'between the sexes' or 'has been developed'
- In the second paragraph on page 2, the phrase 'context-specific data from context-specific gene expression' is redundant
- Page 10 paragraph 2 second to last sentence "worse that" should be "worse than"
- caption of figure 3, "PriorBoost stores" -> "scores"
- page 11 paragraph two "context" is misspelled "conext"
- supplementary figure 6, 'preformaners' -> 'performers' under both tables
- Supplement section 2.3.5, first sentence, "we" should be capitalized

Response to the reviewers':

Reviewer #1 (Remarks to the Author):

The paper explains in a clear way an interesting approach, NetREX, to infer context-specific gene regulatory networks (GRNs) given context-specific gene expression data and an "incomplete" prior GRN. Additionally, a second method is presented, PriorBoost, to assess the informative power of a prior network. The use of statistical learning and graph theory techniques seems very promising. The theoretical foundation of the approach has been carefully designed and carried out, as further explained in the supplemental material.

They provided several benchmarks for their algorithm, and justified the choice of proposed metrics through informative tests. A real effort at interpretation of the results, and cross-checking with other data sources, has been realised. I have some suggestions to support the value of this method.

- It could be interesting to compare the results obtained with NetREX-GENIE3 with existing network tools that i) infer bipartite network (transcription factors regulate their targets) ii) can generate network-based latent representation of transcription factor activity. A few Bioconductor packages such as RTN (Fletcher Et al., 2013; Nature Communications) and CoRegNet (Nicolle et al. 2015; Bioinformatics) as well as others not named here should be included in the comparison and/or discussion.

Reply: Thank you very much for the general appreciation and comments on benchmarking.

With respect to comparing the performance to other method, we did not expected RTN and CoRegNet to outperform Genie3 since in the benchmark comparison DREAM4¹ and DREAM5² challenges that tested over 30 GRN inference methods concluded that Genie3 is the best expression-based method. Tested methods included a group of methods based on the same principles as RTN including the ARACNE approach that RTN implements^{2,3}. Genie3 outperformed all of them. CoRegNet is also an expression-based method but, unlike methods tested in the challenge, can incorporate prior knowledge⁴. However, as shown in Fig. R1 a and d below (which is the same as Fig. 2 in the original manuscript but with RTN and CoRegNet included) both NetREX and Genie3 outperform RTN and CoRegNet. For predicting novel edges, CoRegNet performed better when the prior network is noisy as shown in Fig. R1 b below (10% and 20% of true edges in prior) and Fig. R1. e below (ratio of true to false edges in prior is 0:1).

To the best of our knowledge, Genie3 is the leading expression-based method and Inferelator and MERLIN+P are the leading prior-based methods which motivated our use of these approaches for benchmarking. Given the large number of methods and very thorough comparison performed by DREAM5 (all methods existing at that time)², we think it is reasonable to focus on the winner and newer methods that, similarly to NetREX, utilize a prior. We clarified this in the paper (main text page 2 line 14 and page 3 line 23).

Comparison on various of percentage of true edges in prior networks with fixed number of total edges

Comparison on various of ratio of true to false edges in prior networks with 50% "golden standard" edges

Figure R1. Comparison of network inference methods based on *E. coli* data. **(a)** The performance measured in terms of AUPR as a function of the percentage of true edges in the prior. The total number of edges in the prior networks is fixed and equal to the number of edges in the “gold standard” set. **(b)** Recovery of the novel TF-gene interactions (x axis same as **(a)**). **(c)** PriorBoost scores (x axis same as **(a)**). **(d-f)** Same as (a-c) but when the total number of true edges is fixed and equal to half of the number of “gold standard” edges.

...As the prior network is an important part of the NetRex workflow, it could be interesting to study the robustness of PriorBoost using for example a partially permuted version of the prior network with increasing levels of noise.

Reply: We completely agree that robustness of the PriorBoost score is important. We may not have made sufficient effort to explain the experiments we did to demonstrate the robustness of PriorBoost score in the original manuscript. In fact, we have performed the analysis suggested by the reviewer (Fig. 2 c and f in the main text which is the same as the Fig. R1 c and f in this response) but failed to mention and summarize it in the text. Specifically we varied the percentage of the true edges in the prior while fixing the total number of edges (Fig. R1 c) and additionally we varied the ratio of true to false edges in the prior while fixing the total number of “gold standard” edges. We repeated this procedure 10 times each time randomly selecting a subset of gold standard edges and randomly adding noise to obtain the desired number of edges (Fig. R1 c) or desired ratio of true to false edges (Fig. R1 f). As shown in Fig. R1 c and f, PriorBoost scores become less than 0 when the leading expression-based method (Genie3) outperforms

NetREX, indicating that in these cases the prior network does not help to improve the GRN reconstruction. We now added appropriate summary (main text page 12 line 2).

- The authors should think about availability of an open repository. Also, it would be better to provide a generic implementation of NetRex. The actual version is configured with the demo data and needs Matlab installed. A more detailed explanation should be given about the running time and whether the tool can handle large-scale datasets from RNA-seq and/or Human single cell data (20K cells vs 20K genes). Without an easy-to-use tool this application would be of low significance to the community.

Reply: We agree that making NetREX available in other formats will promote usability. We have now converted our Matlab codes to Python codes and made this available at GitHub: <https://github.com/ncbi/NetREX>.

Both reviewer #1 and #3 mentioned run time and scalability. For computational time and resources that NetREX may utilize, we conducted time and space complexity analysis for NetREX and have included this information in the manuscript. Specifically, NetREX needs to store the expression data and the prior network, therefore, the space complexity of NetREX is $O(NL+NM)$, where N is the number of genes, L is the number of samples, and M is the number of TFs. Based on Algorithm 1 in Supplementary Discussion, the heaviest computation in each iteration of NetREX is to compute the partial derivatives of the objective function, indicating that the time complexity of NetREX in each iteration is $O(NML)$. Therefore, the overall time complexity of NetREX is $O(CNML)$, where C is the number of iterations that NetREX takes in a run. Take the benchmark test on *E. coli* (4511 genes and 334 TFs with 805 samples) for example, reconstructing a single *E. coli* GRN based on fixed parameters, on a computer server with 32 CPUs (2.60GHz) and 132GB memory (NetREX only consume 8GB memory and 8 CPUs), Inferelator and NetREX take around 417 and 814 seconds, respectively, comparing to MERLIN+P and Genie3, both of which take around 20 hours to finish. We have added the above analysis in the main text (page 23 line 10).

Based on the above analysis, NetREX can handle large-scale datasets from RNA-seq and/or Human single cell data. While, the *Drosophila* genome is about 10% of the human genome in length, there are ~17k genes and ~800 TFs in the fly which is reasonably comparable to human data and thus we do not have concerns about scalability. In fact, we plan to use NetREX to build cell-type GRNs from *Drosophila* single cell data.

Minor comments

- abstract : “We validated the predicted GRNs computationally and experimentally ... » This sentence may be confusing as there is no new experiments in the paper to support hypothesis generation. It looks more like cross-checking with other data sources.

Reply: This is a valid complaint, thank you for pointing this out. We have altered this sentence (page 1 line 15) to “We validated the predicted GRNs computationally and performed cross checking using withheld experimental data.”

- Why choosing the *Drosophila* and *E. Coli* DREAM 5 data as they are now quite old datasets and better quality datasets can be obtained now with a better confidence on the differentially expressed genes.

Reply: We used *E. coli* data for benchmarking for several reasons. First, the *E. coli* network is the most complete network and thus provided the best gold standard for comparing the methods ⁵. It has been carefully curated to serve as the benchmark in the DREAM5 challenge ². Furthermore, the fact that *E. coli* DREAM5 data was not up-to-date provided us the opportunity to test how well the networks constructed using older data predict high confidence edges that have been added to RegulonDB ⁵ after completion of the DREAM5 challenge, as we outline in the main text.

As for *Drosophila*, the study we report here is a collaboration between a computational group and a *Drosophila* genetics group making *Drosophila* the organism of choice. Our goal is to construct sex-specific and cell-type-specific networks for this organism that could be used by the *Drosophila* community. As for the dataset, we should clarify that we used here a relatively new and high quality expression dataset ⁶. In fact this data was SPECIFICALLY generated to support this project and has several unique advantages that we did not adequately emphasise in the manuscript. This is a large set of RNA-seq data for both males and females collected systematically by our group that is characterized by large numbers of subtle genetic perturbations. Specifically, the dataset is derived from engineered chromosomal deletions each of which leads to deletion of one of the two copies of a block of genes from different regions. Because all these deletions are heterozygous (and flies are viable and fertile in this state), there are not secondary (and worse) effects due to defects in development. All the flies are morphologically wildtype, but as demonstrated in ⁶ the expression changes caused by these genetic perturbations propagate and dissipate in gene network space, making this an ideal set for expression-based network reconstruction. There are also newer unpublished *Drosophila* datasets available at GSE99574 (96 samples for each of male and female from GSM2647254 to GSM2647349), including several from our group, some of which we use here as validation. We have revised the manuscript to so that it contains more information about the data used (main text page 12 line 12).

- Perhaps consider changing the purple color in figures like Figure 4a by a lighter one (i.e. more readable)

Reply: Thank you for this suggestion, we have changed the dark purple color to light purple.

- Some typos in the manuscript :

page 2: seuxal -> sexual

page 4: we demonstrated that the while (remove "the")

page 7: that infers (add "a") GRN based on

page 7: benefit to starting from -> benefit to start from

page 10: worse that -> worse than

page 11: to test (add "the") consistency

page 12: stores -> scores

page 12: NetREX (add "are") higher than

page 16: details (add "can") be found

page 16: a context-agnostic networks -> a context-agnostic network

page 17: NetREX (add "is") not continuous

page 19: for nonzero elements -> of nonzero elements

Reply: Thanks for pointing out all these typos. We have corrected these and carefully checked for additional ones.

Supplementary - There are a lot of typos and some sentences that should be changed in order to be understood.

Reply: We have fully re-edited the the supplementary materials for spelling, grammar, and syntax.

Reviewer #2 (Remarks to the Author):

Wang et al develop a novel computational approach to derive gene regulatory networks called NetREX. The approach uses prior networks and adds and removes edges to obtain a network that best explains the gene expression data. They also develop a computational approach called PriorBoost that determines if the prior network is useful in estimating gene regulatory networks. The authors validate their approach using simulated data and a “gold standard” gene regulatory network from E. coli. They next used their approaches to construct sex-specific gene regulatory networks from Drosophila data. The approach assumes that a transcription factors is characterized by its activities and gene regulation is due to a linear combination of the activities of transcription factors. The NetREX algorithm starts with a prior and then adds and removes edges, with preference given to network topologies that predict co-expressed genes are co-regulated and transcription factors with similar activities co-regulate the same genes.

To benchmark their new approaches, they first test using simulated data, where a “ground truth” is known. They show that their approach improved network prediction, by examining various levels of “noise” in their simulated expression data. Next they show using the “gold standard” E. coli data that NetREX outperforms other approaches that use priors, if the quality of the prior is good, with sufficient numbers of true edges. The authors next test their approach on Drosophila data, from adult animals harboring different deletion mutations. They generate a prior network based on several different experimental and computational data sets. The PriorBoost approach finds that the prior networks are not a good match for the male data that includes expression data from the germline, but that the prior network is useful on the female data. They proceed removing the testis gene expression and observe good performance. They also test the performance on a separate gene expression data and examine network predictions on non-sex-biased gene expression.

Overall, the benchmarking approaches show that NetREX outperforms other algorithms that predict gene regulatory networks, when there is sufficient prior information. The predicted DSX gene regulatory network from the female data set were significantly supported by the independent biological evidence, providing further proof-of-principle for this new network prediction approach. While the ability to improve prediction of gene networks is a significant step, some questions remain about the data sets used to validate NetREX that the authors need to address.

Reply: We are delighted that the basic outline of the work is clear and appreciate the overall enthusiasm of the reviewer. We address the dataset selection in the detailed responses below.

For the authors to address:

It is not clear why the authors used non-melanogaster data sets for the “second dataset of sex-biased expression”. The GEO number given is GSE80124 and contains Hawaiian Drosophila species data. It is also not clear which data from this data set were analyzed. Given that DSX targets might be different in different species, more rationales as to why they chose a non-melanogaster data set are needed.

Reply: Thank you for catching a serious error! We apologise for providing an incorrect GEO number! The correct GEO number is GSE99574 and we used expression data from only *Drosophila melanogaster* tissues (96 samples for each for male and female from GSM2647254 to GSM2647349) to identify genes expressed in sex-specific way. There are also data from other species in this accession, so we list the specific samples used in the methods. Since we are using this information to evaluate the NetREX network, we deliberately utilise data that was not used for network construction. We have made this clearer in the text (main text page 15 line 19) and expanded the description in Supplementary Discussion II section 2.4.5.

Given that DSX does not regulate gene expression in female germline tissues, does that impact the outcome of the gene regulatory networks constructed, especially when considering sex-biased genes? Including the female germline data and not the male germline data is likely to bias the genes that are called sex-biased in this study. How did the authors deal with this difference in male and female data sets? Did the authors make gene regulatory networks from male and female data sets that are only from somatic tissues? If not, why were those not included?

Reply: The reviewer is absolutely correct about DSX targets. Including the ovary data should decrease performance, because of these indirect non-autonomous effects. Removing both testis and ovary sex-biased expression makes the female and male data more comparable and removes a large number of indirect targets of DSX. Gonad-specific GRNs are a long-term objective of our work, but it is best to leave the gonads out of the current work. This is one of the reasons that we are using tissue-specific (see GSE99574) and single cell (undeposited) libraries going forward. We want to thank the reviewer for making important point that DSX does not regulate gene expression in female germline tissues directly. Indeed, as the reviewer expected, removing female germline data improved our results.

Before we elaborate on details relative to DSX, we would like to respond to the question about the data. For network construction we used the whole fly adult female (for the female network) and male gene expression data (for the male network) including all somatic tissues.

Recognising that NetREX cannot correctly model regulatory program in testis for the reasons explained in the manuscript (given that the logic is a bit strained, we have moved the extended explanation from the main manuscript to Supplementary Discussion II section 2.4.3). We removed from the prior network the genes that are highly expressed in testis and we did not attempt to predict their regulators. However we use it as a nice practical example of how PriorBoost can be used for preventing building GRN based on a too unrelated prior.

Returning to the question related to DSX, following the reviewer's advice, we now also removed female germline data (genes that are highly expressed in ovary) while testing for DSX targets. Technically we identified these genes as follows: first we download RNA-Seq RPKM value dataset from Flybase⁷. Flybase assign genes to bins depending on their expression. "Bin_value" is an integer that ranges from 0 to 6. 0 means a gene has very low expression and 6 means it has extreme high expression. We identified all genes expressed in ovary with "Bin_value" at least 5 and treat them as genes highly expressed in ovary. We recognise, that this procedure removed some housekeeping genes (such as the ribosomal genes) however these genes are not relevant for this test.

After removing these genes we re-benchmarked all competing algorithms, by examining if DSX targets predicted by all methods were enriched in sex-differentially expressed genes. Fig. R2 a and b (below)

show the previous results and the new results. As the reviewer probably expected, the percent of such genes among NetREX's predictions increased significantly after removing the ovary biased expression.

We thank the reviewer for suggestions for this improvement. We moved the whole organism models to the supplement (Supplementary Discussion II: Supplementary Fig. 8 and 9), since it may be useful for researchers interested in gonad expression, but this niche should be filled by follow-up work that is in progress for a different manuscript, that is more focused on the biology rather than the method.

a Female GRN

	Network cutoff (20 TFs / gene)			
	# DSX targets	# DiffExp genes	% DiffExp genes	p-value
NetREX	1900	926	48.74%	2.53E-28
Inferelator	3	0	00.00%	<1.0
MERLIN+P	46	20	43.48%	<0.20
Genie3	411	160	38.93%	<0.4

Background: 7530 genes in the female GRN and 2869 of them are differential expressed (DiffExp) genes. The percentage of DiffExp genes is 38.10%
p-value is obtained from hypergeometric test.

b Female GRN (no ovary)

	Network cutoff (20 TFs / gene)			
	# DSX targets	# DiffExp genes	% DiffExp genes	p-value
NetREX	63	43	68.28%	4.62E-04
Inferelator	3	0	00.00%	<1.0
MERLIN+P	228	94	41.23%	<1.0
Genie3	330	132	40.00%	<1.0

Background: 5916 genes in the female GRN (no ovary) and 2869 of them are differential expressed (DiffExp) genes. The percentage of DiffExp genes is 48.50%
p-value is obtained from hypergeometric test.

Figure R2. Enrichment of predicted DSX targets in genes with sex differentially expressed expression for the female GRNs under the cutoff 20 TFs per gene. (a) Comparison of predicted female GRNs with genes highly expressed in ovary. (b) Comparison of predicted female GRNs without genes highly expressed in ovary. This panel has been added to Fig. 4 d in the main text. Blue ovals circles the percentage of sex differentially expressed genes in the DSX targets predicted with ovary genes and without ovary genes.

The section regarding how non-sex-specific gene regulatory networks were predicted is confusing. What was the prior? Was this also based on the DSX priors?

Reply: We thank the reviewer for pointing out any confusing aspects of this very important baseline information for the paper. For constructing male and female GRNs, the prior was the same (with the caveat that non-expressed genes were not included in the networks which creates slight differences), only the expression data was different. That is, the same network was used as the initial wiring diagram for both sexes.

It is important to note that neither DSX occupancy, nor DSX binding sites were included in the prior. The expression data resulting from direct perturbation of DSX activity was not used either. The modENCODE network constructed by Manolis Kellis' group⁸ is based largely on embryos and tissue culture cells. With the exception of 3 edges, there was no information about DSX in this prior network. Thus, without any

significant prior knowledge of DSX target genes, NetREX predicted them. This a real strength of the work and our failure to point it out clearly is unfortunate.

For validating the predicted GRNs, we evaluated the predicted DSX targets in two ways. For the first, we checked whether the predicted targets overlapped with the targets reported in our previous work ⁹, which was obtained by using DSX occupancy and DSX conserved binding motifs (main text Fig. 4 a-c). For the second, we tested whether those predicted DSX targets were enriched in sex differentially expressed genes (main text Fig. 4 d-e), which were identified based on another independent dataset (in GSE99574, 96 samples from GSM2647254 to GSM2647349). We also looked for the aspects of the sex-specific models that did not predict any differences between the sexes, so there were no non-sex-specific networks, only parts of the sex-specific networks that did not differ between the sexes (main text Fig. 5). We have clarified these issues in the manuscript (page 15 line 4 and line 19).

.... Can the authors provide evidence that there are large sets of DSX targets without sex differences in expression?

Reply: It is important to point out that we made no assumption about existence or not of large sets of DSX targets without sex differences. We assume that DSX targets are enriched in genes that are differentially expressed in male and females and we use it, in addition to other evidence (motif, ChIP-seq main text Fig. 4 a, b and c), as a support for NetREX predictions.

As for the existence of DSX targets without sex differences, in our previous work ⁹ we acutely switched DSX isoforms in adult fatbody and found that only a subset of the occupied genes changed expression, so in fact not all DSX targets are sex-specifically expressed in a given tissue at a given time. This is also now made clear in the text (page 15 line 16).

...What tissue/species were the gene expression data sets made from that were used here? There is too much ambiguity in this section to understand what is being presented.

Reply: For network reconstruction we used whole fly adult female and male gene expression data from ^{6,10} including all somatic tissues (but without dissecting these tissues separately). To make sure that we did not use the same data for network construction and testing, we used a new expression data set from GSE99574 (96 samples from GSM2647254 to GSM2647349) to define sex differential genes for testing (not the Hawaiian species in the incorrect GEO entry in the original manuscript). We used only *D. melanogaster* data. This has been clarified in the manuscript (page 15 line 19).

Minor comments:

The use of the deletion strains is confusing. Can the authors provide more information about why those will show expression variation?

Reply: We did not make this clear in the manuscript and it is important. As outlined in the previous comment, the heterozygous deletions datasets were specifically generated to support this project. Because the perturbation is in blocks of gene copy, not as homozygous mutations in single genes, all the transcriptional effects are perturbed but the underlying GRN is unbroken (the fly is alive and fertile with all its body parts developed, but with significant expression variation propagated through, and ultimately absorbed by, the GRN). These significantly perturbed expression profiles explore the expression space for the whole genome, as collectively essentially all genes show differential expression in at least one deletion. We have extensively rewritten the text to explain this to the reader (main text page 12 line 12).

There are typos in the figures: Figure 1 should be signal and not “singal”. Figure 2-I assume you mean gold standard, not “golden standard”. There are also minor typos throughout the text.

Reply: We have corrected these and other typos in the manuscript.

The authors mention the top 100 NetREX predictions in 4A, but those are not presented in 4A.

Reply: Thanks for pointing out all these typos. We have corrected these and carefully checked for additional ones.

Reviewer #3 (Remarks to the Author):

Summary comments:

This paper describes a method for building context-specific gene regulatory networks given a general specific prior network. It also describes a metric for determining the match between a prior network and a dataset of gene expression data. The authors further define a metric for the quality of a learned network based on PPI networks. Building context-specific networks is very important to the understanding of biological systems specifically in understanding differences between diseased and healthy tissues. Current methods either assume a prior network that was built independent of context or build a network from scratch using data from the system of interest. Using prior networks is an efficient way to both gain confidence in the inferred networks as well as decrease the amount of data required to infer new networks. The metrics provided by this paper for understanding the quality of a prior network and comparing a network to known biology are also important advancements in the field. The authors compare to other methods: two that use priors and another that learns the network directly from the expression data.

While the method seems conceptually interesting, the manuscript could benefit from increased clarity in description of the method in the main text as well as the statistical tests performed in the validation. The authors also should explicitly state the time or space complexity. They are using a method, PALM (proposed proximal alternative linearized maximization), to solve the optimization problem, whose complexity should be stated.

Reply: We are glad that the reviewer has been able to summarize the main points of our work. Reviewer #1 and #2 have also mentioned clarity. We certainly are interested in making the manuscript accessible and show that the work is rigorous, and have made modifications as outlined below.

Major points:

- In figure 1, it seems as though edges between TF's and putative targets will be removed if the one TF is not co-expressed with the other TF's regulating that target. It is possible for a TF to be regulating a gene in an orthogonal way to the other TF's regulating it such that the TF would not be correlated with the other TF's regulating this gene. The authors should clarify if this is an assumption that the method makes.

Reply: Thank you for pointing out the need for clarification. Specifically l₀ elastic net only encourages co-expressed TFs to regulate similar genes but does not remove edges. NetREX only removes edges if it

finds TFs whose activities can better explain the expression of gene(s) than the TFs in the prior network. Then we remove the edges in the prior and add the new edges. We added necessary clarification to the text (main text page 22 line 13) and removed the confusing part out of Fig. 1.

- There is no discussion of the efficiency of the algorithm- is this feasible/fast to run? What is its space complexity?

Reply: We thank the reviewer for this question. Both reviewer #1 and #3 mentioned run time and scalability. We are cognizant of practical aspects of the work, but neglected to mention this in the manuscript. In the revised manuscript we now include complexity analysis (main text page 23 line 10). The time complexity is dominated by the cost of computing the sub-derivative of the objective function (supplementary materials equation (10)). Therefore, the time complexity of each iteration in NetREX is $O(NML)$, where N is the number of genes, L is the number of samples, and M is the number of TFs. The time complexity of NetREX is $O(CNML)$, where C is the number of iterations used in NetREX. The space complexity is $O(NL+NM)$. On *E. coli* data it takes minutes (similarly to Inferelator) as opposed to nearly a day for Genie3 and MERLIN+P (see the response to the Reviewer #1.)

- There is also limited discussion of how much expression data is required for the reported performance. Since expression data is expensive to acquire, it would be useful to have an understanding of the amount needed to obtain the quality reported for this method.

Reply: We thank the reviewer for this important practical question. We used the DREAM5 challenge *E. coli* dataset which has 807 samples and 2066 “gold standard” regulatory edges to explore sample size requirement. We used the “gold standard” *E. coli* network to generate a prior network that is estimated to have 20% of true edges (414 edges out of 2066 “gold standard” edges). The 20% threshold correspond to the threshold for which we start to observe improvements of the prediction made by NetREX over the prediction made by expression only methods. Then we applied NetREX to reconstruct the *E. coli* GRN given the same prior network and randomly selected expression data of various sizes. Specifically, each sample size we randomly selected 10 sets of samples and run NetREX. As shown in Fig. R3, when the sample size is less than 100, the performance of NetREX was quickly improving with the number of samples. After sample size reached ~100, adding additional samples did not have a drastic effect. Interestingly, even with a small number of samples, NetREX provided an improvement over the prior network in terms of AUPR score. We have added this in Supplementary Discussion II section 2.3.4.

Figure R3. Impact of sample size on NetREX performance. (a) The average and standard deviation of AUPR for different sample sizes. (b) The zoom in of (a) between sample size 10 and 100.

- It is unclear how NetREX quantifies statistical significance. The fact that it finds >100 predictions of DSX targets while these other networks get much fewer raises questions about the sensitivity of the method. How is it so much more powerful? (page 13 second to last paragraph).

Reply: For a more transparent comparison, we now use precision - recall curves that do not require any cutoffs. For each method, we rank all predicted DSX targets based on the weights assigned by the respective method and draw precision-recall curves as shown in Fig. R4 below. As the ground truth we use DSX reported in ⁹ based on ChIP-Seq occupancy (peaksum score) and conserved motif scores. We revised Fig. 4 in the text to include this panel Fig. 4 b.

Figure R4. Precision-recall curves for predicting DSX targets for compared methods. The DSX targets predicted by each method are ranked by assigned weights. A high area under the curve corresponds to high precision (low false positive rate) and high recall (low false negative rate). As the ground truth we use DSX targets reported in ⁹ based on ChIP-Seq occupancy and conserved motif scores.

- It would be useful if the authors specifically addressed how this method performs in the case that the prior is consistent with the true network in most cases except one truly differential module of genes. It seems like this method takes advantage of redundancy in biological networks; what happens if some subset of the network is incorrect in a concerted way? This could be very important in inferring networks for disease when using a healthy tissue as a prior and would be of great interest to know if the method handles this type of difference.

Reply: This is a very interesting question. Can the method can handle such non-random error? To answer it we designed a test using simulated data. In a selected module, we rewired the number of true edges from 40% to 100% (Fig. R5 as below). Simultaneously, we varied noise in the expression data. The results are presented on panel b and more detailed description of the test follows. We were actually quite surprised how well NetREX did on this challenging task. We thank the reviewer for suggesting this interesting analysis which we now describe in the revised manuscript (page 8 line 12) and expanding on the details in the Supplementary Discussion II section 2.1.3.

As for the details, we generated the simulated data as follows. First, we randomly generated a GRN between M TFs and N genes (black edges in Fig. R5a). Then, we added a module of n genes and

randomly selected m TFs to regulate the genes in the module. The regulatory interactions between these m TFs and genes in the module from a fully connected bipartite graph (green edges in Fig. R5 a.). Using so constructed true GRN, we generated expression data for $N+n$ genes using the linear model introduced in Supplementary Discussion equation (16) including the addition of expression noise.

To simulate the scenario where “the prior is consistent with the true network in most cases except one truly differential module of genes”, we randomly removed a subset of “true” edges connecting TFs to the genes in the module and randomly reattached them to wrong genes (red edges in Fig. R5a). Then we run NetREX using so perturbed network as the prior and measured Recovery Accuracy (Supplementary Discussion II section 2.1.3) of the true edges leading to the module. We tested how the results depend on two factors: (i) the percentage of the rewired true edges (varied from 40% to 100%) and (ii) added expression noise (Fig. R5 b). NetREX performed very well even in the case when all true edges leading to the module have been removed from the prior. The reason for this high performance can be attributed to the fact that TFs that regulate the module also regulate some genes outside the module allowing NetREX estimate their activities. Then the true edges between TFs and the modules could be recovered by utilizing these, since activities have the capacity to explain the expression of the genes in the module.

Figure R5. Test of NetREX performance under the malicious error model where the prior is consistent with the true network except one truly differential module of genes. (a) Construction of the test data. Rewire $x\%$ edges means that $(1-x)\%$ of true edges leading to the module (green edges) are kept (conserved) and the rest is connected to the wrong genes outside the module (red edges). (b) The performance of NetREX on recovery true edges in terms of Recovery Accuracy under various percentage of rewired edges various level of added expression noise.

Specific points:

The authors allude to the fact that the genie3 network is sometimes closer to the truth than the prior. It would be interesting to see the performance when the genie3 network from the same expression data is used as a prior on NetREX.

Reply: Although AUPR score of Genie3 compares favorably to the prior with 10% of true edges, this is because the AUPR score combines precision and recall. But since Genie3 predicts many edges (much more than the prior), the ratio of true to false edges in the GRN predicted by Genie3 is less than 1:50. Thus, as shown in Fig. 2 d (main text), when the ratio of true to false edges is low, NetREX did not perform well. We expected that Genie3 predicted GRN would be too noisy to be used for NetREX as a

prior and we confirmed this is indeed the case using *E. coli* data (Table R1). This additional de-noising step (e.g. intersecting with additional data to select high confidence edges) would be required before the results of Genie3 can be used as a prior.

Table R1. AUPR and AUROC score for the Genie3 predicted GRN and the GRN predicted by NetREX using the Genie3 prediction as the prior.

Methods	AUPR	AUROC
Genie3	0.0899	0.6499
Genie3+NetREX	0.0595	0.6508

1. Specific comments on the tests of the prediction of novel edges from RegulonDB:

- Were the new interactions they tested from regulonDB all in one module? Were they at all context specific? Some information about which edges these were and which ones were recovered could strengthen the argument that NetREX is specifically adept at identifying context-specific networks.

Reply: To answer this question we show below (Fig. R6) 230 novel edges as a network coloring the modules that enriched in a biological process (GO enrichment analysis under $FDR < 0.001$). From the figure, we could see that many of the interactions are from the same functional modules but most of the edges are not.

However, we do not see how this information can be used to strengthen the argument that NetREX is specifically adopted at identifying context-specific networks. We need to clarify that context is provided by expression data. In particular, starting from the same prior but different context specific expression data, we can optimize this network in a context depending way to explain each of the provided contexts (in this paper male/female). However for *E. coli* we have one expression dataset so one context. Therefore, *E. coli* network is used as a test of the quality of our optimisation method for a specific context. We now clarified this aspect in text (page 3 line 6 and line 9).

Figure R6. The network of 230 novel *E. coli* edges. Modules that enriched in a certain biological process (GO enrichment analysis under FDR < 0.001) are colored.

- In figure 2b, the authors should clarify how they calculated the p-value and what it is quantifying. Based on the supplementary tables (5 and 6) it seems that the hypergeometric test is testing for enrichment of

novel edges in the total number of edges found by the algorithm, but these bar plots could also be interpreted as the enrichment of novel edges found by the algorithm in the set of total novel edges. A clarification of this difference would be useful to include.

Reply: We used the hypergeometric test for enrichment of novel edges in the set of total novel edges found by the algorithm. We have clarified this in the main text (page 10 line 13).

- Tables 5/6 in the supplement indicate that there is a large difference in the total number of predicted edges between the three algorithms. While the AUPR metric addresses the question of false positive edges, including raw numbers on true vs. false positives in tables 5 and 6 as well as a comment on the differences between the numbers of edges found would add clarity.

Reply: As suggested by the reviewer, we now replaced in Table 5 and 6 averages with the raw numbers. Specifically, we show number true and false positives (means \pm standard deviation). As the reviewer can now appreciate, NetREX predicted fewer False Positive (FP) edges but comparable number True Positive (TF) edges relative to other methods explaining the favorable differences in p-values.

Table 5: The comparison of the methods based on the ability to identify novel interactions that were not used in the DREAM5 challenge as a function of quality of the prior network where the prior quality is measured as the percentage of true edges.

true edge percentage	Inferelator			NetREX			MERLIN_P		
	# TP	# FP	# Unique	# TP	# FP	# Unique	# TP	# FP	# Unique
10%	5 \pm 2.82	15,094 \pm 203.5	9	4 \pm 2.97	3,710 \pm 80.4	8	24 \pm 2.89	49,033 \pm 166.6	28
20%	9 \pm 3.13	14,606 \pm 337.4	14	5 \pm 2.26	3,422 \pm 80.7	21	23 \pm 1.53	48,762 \pm 325.8	27
30%	8 \pm 2.02	14,480 \pm 277.5	19	7 \pm 1.90	3,078 \pm 51.1	22	25 \pm 0.58	49,066 \pm 302.1	26
40%	10 \pm 1.32	14,326 \pm 270.0	18	10 \pm 3.51	2,860 \pm 96.5	26	25 \pm 1.73	48,883 \pm 420.0	28
50%	11 \pm 3.07	14,227 \pm 210.7	17	11 \pm 2.25	2,613 \pm 55.1	25	23 \pm 2.08	48,648 \pm 137.5	23
60%	13 \pm 3.40	14,187 \pm 348.4	19	14 \pm 2.45	2,389 \pm 64.7	25	25 \pm 1.53	48,733 \pm 58.9	28
70%	14 \pm 2.28	13,948 \pm 190.7	18	19 \pm 1.78	2,256 \pm 63.4	29	23 \pm 1.15	48,417 \pm 58.9	26
80%	13 \pm 1.06	14,063 \pm 249.4	19	20 \pm 3.02	2,151 \pm 63.4	33	25 \pm 1.15	48,546 \pm 573.5	28
90%	15 \pm 1.58	13,833 \pm 177.3	18	21 \pm 2.64	1,994 \pm 59.5	30	25 \pm 1.25	48,446 \pm 387.6	28

Elements in the table are means \pm standard deviation. TF and FP are True Positives and False Positives.

Unique is the number of identified unique novel edges over 10 runs of the methods starting with randomly selected priors.

Table 6: The comparison of the methods based on the ability to identify novel interactions that were not used in the DREAM5 challenge as a function of quality of the prior network where the prior quality is measured as the ratio of true to false edges.

ratio of true to false	Inferelator			NetREX			MERLIN-P		
	# TP	# FP	# Unique	# TP	# FP	# Unique	# TP	# FP	# Unique
1:0	16 ± 2.50	15,052 ± 124.8	25	19 ± 1.91	3,014 ± 56.4	26	25 ± 1.91	46,493 ± 481.5	28
1:2	12 ± 4.67	15,085 ± 441.3	20	11 ± 2.36	5,088 ± 78.3	34	24 ± 0.00	47,033 ± 413.1	26
1:5	9 ± 4.48	15,292 ± 269.6	18	11 ± 2.27	8,187 ± 85.4	49	23 ± 0.58	46,751 ± 279.9	24
1:10	8 ± 3.21	15,252 ± 378.3	12	12 ± 3.41	13,363 ± 104.4	53	24 ± 0.58	47,187 ± 274.5	26
0:1	5 ± 4.81	15,454 ± 317.4	11	1 ± 1.96	3,032 ± 65.4	10	23 ± 2.08	46,607 ± 272.5	24

Elements in the table are means ± standard deviation. TP and FP are True Positives and False Positives.

Unique is the number of identified unique novel edges over 10 runs of the methods starting with randomly selected priors.

- The column names in tables 5 and 6 (“# avg. novel” and “# avg. overall”) should be explained more clearly (i.e. how many tests went into this average? What is the spread of the average?

Reply: As suggested we replaced # avg. novel and # avg. overall by raw numbers reporting # true and #false positives (means ± standard deviation) in Table 5 and 6. We clarified that each test was performed 10 times.

...Are the same novel and overall edges found in each test of the same algorithm?)

Reply: Not necessarily. Since a very noisy prior is selected randomly, it is different at each test leading to quite different predictions. However when the prior is not very noisy, then edges found in each test are largely overlapping. To see this we included the total number of uniquely identified edges over 10 runs of the methods starting with randomly selected priors in Supplementary Discussion II: Table 5 and Table 6.

- Referencing the section of the supplementary information (tables 5 and 6) containing the results of the test of recovering E.coli new interactions on page 10 in the second to last paragraph would increase readability.

Reply: We have revised this part as suggested (main text page 10 line 15).

2. Specific comments on recovering DSX targets from Drosophila data:

- While interesting, the description of the motivation to remove testes-specific genes might be off topic for this paper (bottom of page 12-page 13).

Reply: We agree that this is a bit off topic so we moved most of the details of this description to the supplement (Supplementary Discussion II: section 2.4.3) leaving only a general comment that there are profound differences in regulation of testis-specific genes explaining the need to remove these genes.

- The authors should clarify if the DSX targets are expected to be different for males and females and the degree to which they are different. Specifically, when reporting the results for enrichment of DSX targets identified for males and females, the authors should indicate how many of the identified targets were truly context-specific and how many were targets found in both male and female networks.

Reply: We thank the reviewer for pointing the need for a clarification. We published an entire paper on DSX context-dependency ⁹, which we don't want to repeat here, but we indeed do need to clarify this in the main text. Males and females express a different DSX isoform. These isoforms often bind to the same sites in both sexes. However, even if the edge is the same in the male and female network the target genes can still be differentially regulated (up versus down, for example). Thus truly context-specific gene regulation might still be accompanied by the same edge in both male and female making it difficult to perform a test as the reviewer suggested. However we designed a different test that demonstrates the power of NetREX to capture context specific regulation. Using an independent dataset we classified genes based on whether or not they are differentially expressed between male and female adults. We reasoned that, accounting for imperfections in the GRN network prediction, we still should expect that genes that are not differentially expressed between male and female have higher similarity of regulatory interactions than genes that show sex specific expression. This is indeed what we found. The results of the test are demonstrated in the revised Fig. 5 in the manuscript. Importantly, this property can be observed for networks constructed by NetREX, but not for other networks we tested. We added this information in the text (page 17 line 16).

In direct answer to the question, DSX has 91 and 1077 targets in male and female networks respectively. There are only 19 targets existing in both networks and 6 of them are sex-differentially expressed genes.

The authors mention that the prior networks used for male and female were different (supplementary table 7), but in supplementary note section 2.3.2 the authors specify that the same prior network was used for both. The authors should clarify which network was used for each step. For the steps using different priors, the source of these networks should be clarified as well as the motivation for using different networks. The authors should also clarify which of these TF-target pairs are included in the prior.

Reply: We thank the reviewer for pointing out the need to clarify this important issue. The prior networks for female and male are basically the same and correspond to the network predicted in ⁸. However, genes that are not expressed are removed from the prior. Since the set of non-expressed genes in females and males is not exactly the same, this introduces a subtle difference between the two priors. We have clarified the above point in the manuscript (page 13 line 4).

Is there a difference in the number of edges NetREX must add/remove from the prior for the male and female networks? Are the priors closer to the male or female context specific network? In supplementary section 2.3.5 the authors state that the NetREX recovers more overlaps than other methods, and knowing the number of these overlaps that were included in the priors would help in understanding the advantages of this method.

Reply: There are 505,548 and 293,458 edges predicted by NetREX for female and male (no testis) GRNs. We found that 149,462 edges are common to the female and male GRNs 136,404 of the common edges are included in the prior and 13,058 edges are newly predicted. We added these edge statistics to the text (page 17 line 2).

- The methods' description for highly expressed genes in testis is vague. Where did the testis-specific gene list come from?

Reply: We used the quantification of tissue specific expression from modENCODE as summarized in FlyBase. Flybase assign genes to bins depending on their expression in given tissue. "Bin_value" is an integer that ranges from 0 to 6 where 0 means that a gene has very low expression and 6 means it has extreme high expression. We identified all genes expressed in testis with "Bin_value" a least 5 and treat them as genes highly expressed in testis. This simple method suffices to demonstrate our point. We have added this information (page 26 line 15).

- The hypergeometric tests in figure 4a are unclear. The background should be specified more clearly as the interpretation of the p-value of 1 for the prior and the Inferrelator method comparisons is confusing without specification of how the test was run. In supplementary table 11, the same values for predicted and verified genes as in figure 4a are listed for the prior network, but a different p-value is stated. A description of the background would help clarify this difference.

Reply: The background of the hypergeometric test is that there are 3225 DSX targets reported in ⁹ within 5916 genes. The p-values in Fig. 4 a (main text) and table 11 (Supplementary Discussion II) are results obtained from the predicted female specific GRN and male specific GRN, respectively. The difference between Fig. 4 a and table 11 (Supplementary Discussion II) is because they correspond to different sexes. As per suggestion of reviewer #2 in the revised version we omitted genes highly expressed in ovary or testis in this comparison. We have added this information in Fig. 4 a.

- When testing for the enrichment of DSX targets in differentially expressed genes, it is conventional to use Gene Set Enrichment Analysis as it takes into account the rankings of the differential expression of the genes. The authors should justify why the hypergeometric test is sufficient for this.

Reply: As per reviewer advice, we have also used Gene Set Enrichment Analysis (WebGestalt ¹¹) to test the enrichment of DSX targets in differentially expressed genes. The result is in the following Table R2, showing that only the prediction of NetREX is enrichment under an FDR 0.001 cutoff. We added these results to the Fig. 4 e.

Table R2. Enrichment of DSX targets in sex differentially expressed genes by GSEA.

Data	Method	NetREX	MERLIN+P	Inferelator	Genie3
Female GRN	GESA(FDR<0.001)	Yes (<2.2e-16)	No	No	No

p-value 2.2e-16 is the smallest p-value that WebGestalt could return.

- The authors should clarify what the cutoff '30 TFs based on edge weights means' is referring to in supplement section 2.3.4 and in the sections beyond that.

Reply: As mentioned earlier in this response, all compared methods return ranked lists of predictions. They might predict different number of regulators for each gene. To fairly compare those GRNs we take for each method the k-best predictions for each gene. In the Fig. 4 we show k=20 and in supplement k=30 and 40. Results are robust to cutoff values used. We included this clarification in the text (page 16 line 10).

- For the DESeq2 outputs in supplementary section 2.3.4, the authors state that the filtering for differentially expressed genes is done via a log2fold change cutoff. Usually, a cutoff for adjusted p-values is performed on the outputs of DESeq2 as well, does performing this cutoff change the outputs significantly?

Reply: We thank the reviewer for this suggestion. As suggested in ¹², we now use both the log2fold change and adjusted p-value to determine the differentially expressed genes (details in Supplementary Discussion II section 2.3.5). The specific cutoffs we used are log2fold (>2.0) and adjusted p-values (<1E-3). The more rigorous selection improved our results (Fig. 4 d and e).

- Some indication of the spread of the Jaccard index should be included for the averages in figure 5.

Reply: We thank the reviewer for the suggestion for improvement. We now added the spread and in (updated) Fig. 5 we added the standard deviation of the Jaccard index.

Typos:

- On page 2 in the top paragraph, the prose “accommodate regulatory program reality” is ambiguous
- on page 2 in the second paragraph there should be no comma after ‘between the sexes’ or ‘has been developed’

- In the second paragraph on page 2, the phrase ‘context-specific data from context-specific gene expression’ is redundant

-Page 10 paragraph 2 second to last sentence “worse that” should be “worse than”

- caption of figure 3, “PriorBoost stores” -> “scores”

-page 11 paragraph two “context” is misspelled “conext”

- supplementary figure 6, ‘preformaners’ -> ‘performers’ under both tables

- Supplement section 2.3.5, first sentence, “we” should be capitalized

Reply: We have corrected these typos and carefully proofread the document to spot any other issues.

Response References

1. Marbach, D. *et al.* Revealing strengths and weaknesses of methods for gene network inference. *Proc. Natl. Acad. Sci. U. S. A.* **107**, 6286–6291 (2010).
2. Marbach, D. *et al.* Wisdom of crowds for robust gene network inference. *Nat. Methods* **9**, 796–804 (2012).
3. Fletcher, R. J., Jr *et al.* Network modularity reveals critical scales for connectivity in ecology and

- evolution. *Nat. Commun.* **4**, 2572 (2013).
4. Nicolle, R., Radvanyi, F. & Elati, M. CoRegNet: reconstruction and integrated analysis of co-regulatory networks. *Bioinformatics* **31**, 3066–3068 (2015).
 5. Méndez-Cruz, C.-F. *et al.* First steps in automatic summarization of transcription factor properties for RegulonDB: classification of sentences about structural domains and regulated processes. *Database* **2017**, (2017).
 6. Lee, H. *et al.* Effects of Gene Dose, Chromatin, and Network Topology on Expression in *Drosophila melanogaster*. *PLoS Genet.* **12**, e1006295 (2016).
 7. Gramates, L. S. *et al.* FlyBase at 25: looking to the future. *Nucleic Acids Res.* **45**, D663–D671 (2017).
 8. Marbach, D. *et al.* Predictive regulatory models in *Drosophila melanogaster* by integrative inference of transcriptional networks. *Genome Res.* **22**, 1334–1349 (2012).
 9. Clough, E. *et al.* Sex- and tissue-specific functions of *Drosophila* doublesex transcription factor target genes. *Dev. Cell* **31**, 761–773 (2014).
 10. Ryder, E. *et al.* The DrosDel deletion collection: a *Drosophila* genomewide chromosomal deficiency resource. *Genetics* **177**, 615–629 (2007).
 11. Wang, J., Vasaikar, S., Shi, Z., Greer, M. & Zhang, B. WebGestalt 2017: a more comprehensive, powerful, flexible and interactive gene set enrichment analysis toolkit. *Nucleic Acids Res.* **45**, W130–W137 (2017).
 12. Dalman, M. R., Deeter, A., Nimishakavi, G. & Duan, Z.-H. Fold change and p-value cutoffs significantly alter microarray interpretations. *BMC Bioinformatics* **13 Suppl 2**, S11 (2012).

REVIEWERS' COMMENTS:

Reviewer #1 (Remarks to the Author):

In this revised version of the manuscript, the authors addressed most of my remarks about the previous version.

My concern ("minor") still is about comparisons with other methodologies (Fig R1). The authors show in their response letter that CoRegNet performed better (for predicting novel edges) when the prior network is noisy. I think that these results should be included in the paper (Fig 2) as it's the only case where NetREX is outperformed by another approach (Inferelator, MERLIN+P, GENIE3).

If I am aware about article size, I think that given how NetREX handle the « undersampling problem » (Fig R3) is important results and I suggest to be considered in Fig 2 (performance evaluation and comparison with Insilco DREAM5 dataset).

Reviewer #2 (Remarks to the Author):

The changes made by the authors address all the points raised in the review. The changes to the text and analysis strengthen the paper overall.

I recommend that the authors include the lists of fly genes identified by their gene regulatory network analysis as supplemental data tables, in addition to the having the results in the figures. I did not see those data tables as part of the submitted manuscript.

The text can still be improved by a careful copy edit. I also encourage the authors to think about their different audiences--computational/statistical biologists and experimental biologist--in terms of making the writing clear. It seems the authors used the supplemental discussion section to help in this regard. Will this be obvious to the reader and if not can they think of ways to point their reader to sections that would helpful. I think the types of additions to the text made during review really helped clarify rationales.

Reviewer #3 (Remarks to the Author):

The method for calculating the p-values in figure 4a is still not clear.

It seems like the method for calculating the p-values would be (in the example of the prior or Inferelator networks) for the probability of selecting 3 DSX targets in a sample of 3 genes from a population with 5916 genes 3225 of which are DSX targets, but this does not yield a p-value of 1 using the hypergeometric test on the upper tail. Can the authors clarify the method for calculating this p-value again?

Other than that, they addressed all the comments, and I think everything looks good!

Response to the reviewers':

Reviewer #1 (Remarks to the Author):

In this revised version of the manuscript, the authors addressed most of my remarks about the previous version.

Reply: We thank the reviewer for the encouraging comments.

My concern ("minor") still is about comparisons with other methodologies (Fig R1). The authors show in their response letter that CoRegNet performed better (for predicting novel edges) when the prior network is noisy. I think that these results should be included in the paper (Fig 2) as it's the only case where NetREX is outperformed by another approach (Inferelator, MERLIN+P, GENIE3).

Reply: We have added CoRegNet in Fig 2 in main text and emphasised that CoRegNet outperforms others for low quality prior in predicting novel edges in main text (page 7 line 11).

If I am aware about article size, I think that given how NetREX handle the « undersampling problem » (Fig R3) is important results and I suggest to be considered in Fig 2 (performance evaluation and comparison with Insilco DREAM5 dataset).

Reply: We are pleased that the reviewer recognises the importance of this result. In response to his/her suggestion we added a short statement of this result to the main text (page 7 line 2) but dues to space limitation we had to keep a longer description in the supplementary text.

Reviewer #2 (Remarks to the Author):

The changes made by the authors address all the points raised in the review. The changes to the text and analysis strengthen the paper overall.

Reply: We thank the reviewer for the encouraging comments.

I recommend that the authors include the lists of fly genes identified by their gene regulatory network analysis as supplemental data tables, in addition to the having the results in the figures. I did not see those data tables as part of the submitted manuscript.

Reply: We have provided the female-specific and male-specific GRNs predicted by NetREX in the Supplementary Data 1 and Supplementary Data 2 and mentioned the availability of the data in the main text (page 9 line 22).

The text can still be improved by a careful copy edit. I also encourage the authors to think about their different audiences--computational/statistical biologists and experimental biologist--in terms of making the writing clear. It seems the authors used the supplemental discussion section to help in this regard. Will this be obvious to the reader and if not can they think of ways to point their reader to sections that would help. I think the types of additions to the text made during review really helped clarify rationales.

Reply: Thank you for this positive comments and we thank in this reviewer for his/her previous constructive comments that helped us to make the paper more accessible to diverse audience. The paper combines sophisticated mathematical techniques and biological results making it challenging to write. We proofread the text to see if additional cross-references (at 17 different places) within text and supplement would be helpful.

Reviewer #3 (Remarks to the Author):

The method for calculating the p-values in figure 4a is still not clear.

It seems like the method for calculating the p-values would be (in the example of the prior or Inferelator networks) for the probability of selecting 3 DSX targets in a sample of 3 genes from a population with 5916 genes 3225 of which are DSX targets, but this does not yield a p-value of 1 using the hypergeometric test on the upper tail. Can the authors clarify the method for calculating this p-value again?

Reply: We thank the reviewer for the insightful comments. Using the same example the reviewer mentioned, the p-value is the probability of selecting at least 3 DSX targets in a sample of 3 genes from a population with 5916 genes 3225 of which are DSX targets. p-value “1” is a typo for this, it should be 0.162. We have checked the rest of the table in Fig. 4a and corrected the typos.

Other than that, they addressed all the comments, and I think everything looks good!

Reply: We thank the reviewer for the encouraging comments.